# Correlation functions and characteristic lengthscales in flat band superconductors

M. Thumin[*] and G. Bouzerar[†]

*Université Grenoble Alpes, CNRS, Institut NEEL, F-38042 Grenoble, France*
(Dated: 11 septembre 2024)

The possibility of an unconventional form of high temperature superconductivity in flat band (FB) material does not cease to challenge our understanding of the physics in correlated systems. Recently, it has been argued that the coherence length in FB compounds could be decomposed into a conventional part of BCS type and a geometric contribution which characterises the FB eigenstates, the quantum metric. Here, we propose to address this issue in various FB systems and discuss whether the extracted characteristic lengthscales such as the size of the Cooper pairs obey this conjecture. It is found that the relevant lengthscales are less than one lattice spacing, weakly sensitive to the strength of the electron-electron interaction, and more importantly disconnected from the quantum metric.

## INTRODUCTION

Over the past ten years we are witnessing a rapidly growing interest for the physics in dispersion-less bands [1–8]. In flat band (FB) compounds, because the width of these bands is extremely narrow, the Coulomb energy is left as the unique relevant energy scale. This places naturally these systems in the class of highly correlated materials and opens the access to exotic and unexpected physical phenomena and quantum phases. Undeniably, one of the most striking feature is the possibility of high critical temperature superconductivity (SC) in compounds where the Fermi velocity vanishes [9–18]. In contrast to conventional superconductivity, this unconventional form of superconductivity is of inter-band nature. In other words, the superfluid weight is controlled by the off-diagonal matrix elements (in terms of band index) of the current operator, and the diagonal contribution (conventional contribution) vanishes or is negligible. The superconductivity in FBs is characterised by a geometrical quantity known as the quantum metric (QM). The QM is connected to the real part of the quantum geometric tensor [19, 20] and its square root provides a measure of the typical spread of the FB Bloch eigenstates. So far, the unique experimental realisation of such an unusual form of superconductivity is very likely the one that has been observed in twisted bilayer of graphene (Moiré) in the vicinity of magic angles [8, 21–26].

It is well known that in conventional BCS systems where the superconductivity is of intra-band nature [27, 28], the coherence length $\xi_c$ is given by $\xi_{BCS} = \frac{\hbar v_F}{\Delta}$ where $v_F$ and $\Delta$ are respectively the Fermi velocity and superconducting gap or pairing amplitude. We recall that $\xi_c$ measures the size of the Cooper pair in real space. Since, in the BCS regime (weak coupling) the superconductivity gap is exponentially small, $\xi_c$ is often extremely large, hence Cooper pairs are highly overlapping with each other. On the other hand, in the strong coupling regime the Cooper pairs can be assimilated to tightly bound non-overlapping composite bosons which at low temperature leads to the well known Bose Einstein condensation phenomenon (BEC)[29, 30].

A natural question arises : what about the case of FB superconductors ? Recently, it has been argued that the coherence length in these systems has two contributions, the first is of conventional type and the other is purely geometric in nature [31, 32]. More precisely, it is claimed that the coherence length can be expressed as $\xi_c = \sqrt{\xi_{BCS}^2 + \langle g \rangle}$ where $\langle g \rangle$ is the average of the QM. Hence, if the band is rigorously flat the first term vanishes.

The purpose of the present study is to address this issue in several FB lattices and discuss our findings in connection with these predictions and with the existing literature. More precisely, we propose to consider four different systems, three of them are one dimensional and the last one is two dimensional : the stub lattice, the sawtooth chain, the Creutz ladder and the $\chi-$lattice. These models and their respective dispersions (in the non interacting case) are depicted in Fig.1. Notice that the $\chi$-Lattice has been originally introduced in Ref. [33]. However, since no specific name has been attributed to this peculiar model,"$\chi-$lattice" has been chosen. In this system, the range of the extended hoppings is controlled by a single parameter ($\chi$) as it will become more explicit in the next paragraph. The choice of these four different systems is motivated by several intentions. It allows to estimate the impact of (i) the bipartite character of the lattice, (ii) the tunability of the quantum metric, (iii) the absence of dispersive bands in the spectrum, (iv) and last the lattice dimension.

## THEORY AND METHODS

Electrons are described by the attractive Hubbard model which reads,

$$\hat{H} = \sum_{i\lambda, j\eta, \sigma} t_{ij}^{\lambda\eta} \, \hat{c}_{i\lambda,\sigma}^{\dagger} \hat{c}_{j\eta,\sigma} - \mu \hat{N} - |U| \sum_{i\lambda} \hat{n}_{i\lambda,\uparrow} \hat{n}_{i\lambda,\downarrow}, \quad (1)$$

where $\hat{c}_{i\lambda,\sigma}^{\dagger}$ creates an electron of spin $\sigma$ at site $\mathbf{r}_{i\lambda}$, $i$ being the cell index and $\lambda$ the orbital index ranging from 1 to

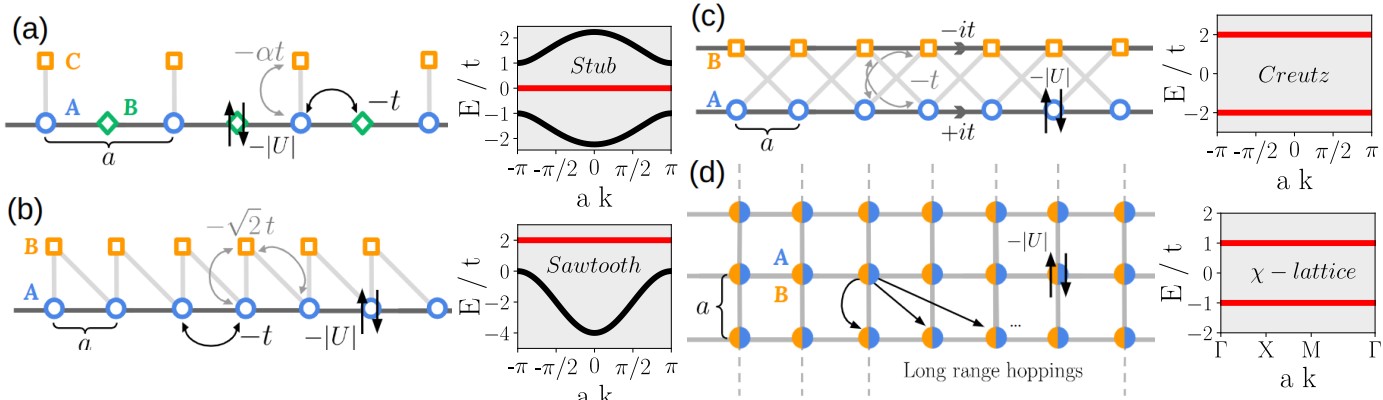

FIGURE 1. Schematic representation of **(a)** the stub lattice, **(b)** the sawtooth chain, **(c)** the Creutz ladder and **(d)** the two-dimensionnal $\chi$-lattice. Their respective dispersions, in the non interacting case, are depicted in the panels having a grey background. The hoppings and the on-site Hubbard attractive interaction term are depicted in the figure. In the case of the $\chi$-Lattice (two orbitals A and B per site) the hoppings are long range (see main text).

$n_{orb}$. $\hat{N} = \sum_{i\lambda,\sigma} \hat{n}_{i\lambda,\sigma}$, $\mu$ is the chemical potential and $|U|$ is the strength of the on-site attractive electron-electron interaction. The hoppings are very short ranged in the stub lattice, the sawtooth chain and the Creutz ladder as depicted in Fig.1. On the other hand, in the $\chi$-Lattice the situation differs, the hoppings are long-ranged, restricted to $(A, B)$-pairs, and given by $t_{ij}^{AB} = -\frac{t}{N_c}\sum_{\mathbf{k}} e^{i\mathbf{k}\cdot\mathbf{r}} e^{i\gamma_{\mathbf{k}}}$ where $\gamma_{\mathbf{k}} = \chi(\cos(k_x a) + \cos(k_y a))$, $\mathbf{r} = \mathbf{r}_j - \mathbf{r}_i$, and $N_c$ being the number of unit cells. The parameter $\chi$ controls both the range of the hoppings and the QM which is given by $\langle g \rangle = \chi^2 a^2/8$ [34].

In this work, we treat the interaction term within the Bogoliubov de Gennes (BdG) approach which consists in the following decoupling scheme,

$$\hat{n}_{i\lambda,\uparrow}\hat{n}_{i\lambda,\downarrow} \overset{BdG}{\simeq} \langle \hat{n}_{i\lambda,\downarrow}\rangle\hat{n}_{i\lambda,\uparrow} + \langle \hat{n}_{i\lambda,\uparrow}\rangle\hat{n}_{i\lambda,\downarrow}$$
$$+ \frac{\Delta_{i\lambda}}{|U|}\hat{c}_{i\lambda,\uparrow}^\dagger\hat{c}_{i\lambda,\downarrow}^\dagger + \frac{\Delta_{i\lambda}^*}{|U|}\hat{c}_{i\lambda,\downarrow}\hat{c}_{i\lambda,\uparrow}, \quad (2)$$

where the self-consistent parameters $\langle \hat{n}_{i\lambda,\sigma}\rangle$ and $\Delta_\lambda = -|U|\langle \hat{c}_{i\lambda\downarrow}\hat{c}_{i\lambda\uparrow}\rangle$ are respectively the orbital dependent occupations and pairings. $\langle \ldots \rangle$ corresponds to the grand canonical average. Notice, that the total carrier density is defined as $n = N_e/N_c$, where $N_e$ is the total number of electrons, hence $n$ varies from 0 to $2\,n_{orb}$.

Before we discuss our calculations, we propose to provide some arguments that justify that our approach is meaningful. We first start with the shortcomings. It is well established that the BdG Hamiltonian being quadratic, it is inappropriate to calculate reliably two particles correlation functions (CFs) such as the pairing-pairing correlation function $f_P(\mathbf{r}_i - \mathbf{r}_j) = \langle \hat{\Pi}_i^\dagger\hat{\Pi}_j\rangle$ where the on-site pairing operator (s-wave) $\hat{\Pi}_i^\dagger = \hat{c}_{i\uparrow}^\dagger\hat{c}_{i\downarrow}^\dagger$. In the case of the attractive Hubbard model in two dimensional systems, one expects the correlation function $f_P(\mathbf{r})$ to decay algebraically with a $T$-dependent power for $T < T_{BKT}$, and exponentially when $T > T_{BKT}$, where $T_{BKT}$ is

the Berezinskii-Kosterlitz-Thouless transition temperature [35–37]. On the other hand, the one-particle CF of the form $f_{sp}^\sigma(\mathbf{r}_i - \mathbf{r}_j) = \langle \hat{c}_{i\sigma}^\dagger\hat{c}_{j\sigma}\rangle$ always decays exponentially in the superconducting phase. Mean Field theory such as the BdG approach can not describe the change of behaviour of $f_P(\mathbf{r})$ across the BKT transition, since through Wick's theorem two-particles CFs reduce to products of one-particle CFs only. However, in FB systems, one expects the single particle CFs to be well captured within the BdG theory. For instance, it has been shown, that the local occupations, the pairings and the superfluid weight calculated by the numerically unbiased DMRG are in excellent agreement with the mean field values in the Creutz ladder and in the sawtooth chain [12, 38]. It should be emphasised that the agreement found concerns both the weak and the strong coupling regime. In what follows it will be shown that it is as well the case for correlations functions.

To study the characteristic lengthscales in the superconductivity phase at $T = 0$, we define the normal and anomalous CFs,

$$G_{\lambda\eta}(\mathbf{r}) = \langle \hat{c}_{i\lambda,\sigma}^\dagger\hat{c}_{j\eta,\sigma}\rangle, \quad (3)$$
$$K_{\lambda\eta}(\mathbf{r}) = \langle \hat{c}_{i\lambda,\uparrow}\hat{c}_{j\eta,\downarrow}\rangle, \quad (4)$$

where the index $i$ (respectively $j$) refers to the unit cell position $\mathbf{r}_i$ (respectively $\mathbf{r}_j$), $\lambda$ (resp. $\eta$) labels the orbitals, and $\mathbf{r} = \mathbf{r}_j - \mathbf{r}_i$. Here, the spin index $\sigma = \uparrow, \downarrow$ is irrelevant, the superconductivity phase being non magnetic. The CF $K_{\lambda\eta}$ is particularly of interest since it allows the extraction of the Cooper pair size. Indeed, in the case of a single one dimensional dispersive band problem (conventional SC) it can be shown analytically that $K_{\lambda\lambda}(\mathbf{r}) \simeq \frac{1}{\sqrt{|\mathbf{r}|}}e^{-|\mathbf{r}|/\xi_{BCS}}$ for $|\mathbf{r}| \to \infty$ as addressed in the next paragraph.

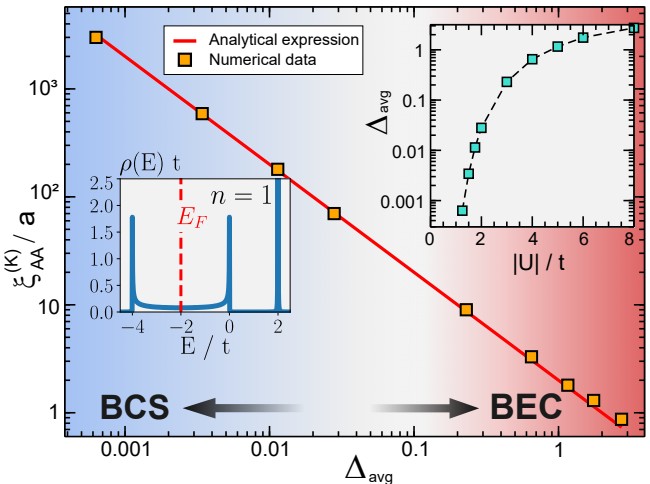

FIGURE 2. $\xi_{AA}^{(K)}$ as a function of the averaged pairing $\Delta_{avg}$ in the quarter filled sawtooth chain. The red thick line is the BCS formula $\frac{\hbar v_F}{\Delta_{avg}}$ where $\hbar v_F = 2at$. The first inset (top right) shows the correspondence between $|U|$ and $\Delta_{avg}$ and the other one illustrates the density of states for $|U| = 0$, with $E_F = -2t$ for the quarter filling. The BCS regime corresponds to $\xi_{AA}^{(K)} \gg a$ and BEC to $\xi_{AA}^{(K)} \le a$.

## RESULTS AND DISCUSSIONS

### Coherence length in dispersive bands

Before we discuss in details the case where the Fermi energy coincides with that of the FB, it is interesting to analyse the situation where it is located inside the dispersive bands. To illustrate this scenario, we consider the quarter filled sawtooth chain. This density corresponds to the half-filling of the lower dispersive band.

In Fig.2, $\xi_{AA}^{(K)}$ is plotted as a function of the averaged pairing $\Delta_{avg}$ in the quarter filled sawtooth chain where $\Delta_{avg} = \frac{1}{2}(\Delta_A + \Delta_B)$ (A and B sites are inequivalent). This characteristic lengthscale is obtained from a fit of the form $\frac{1}{\sqrt{|r|}}e^{-|r|/\xi_{AA}^{(K)}}$ of the long distance behaviour of the anomalous CF $K_{AA}(\mathbf{r})$. The BCS-like expression (red thick line in the figure) is defined as $\frac{\hbar v_F}{\Delta_{avg}}$. Here the Fermi velocity $v_F = \frac{2a\,t}{\hbar}\sin(k_F a)$ where $k_F a = \frac{\pi}{2}$ for the quarter filled sawtooth chain. It is striking to see that the excellent agreement found between the numerical data and the BCS expression is not restricted to the weak coupling regime ($\Delta_{avg} \ll t$). Indeed, remarkably the agreement is obtained for values of the average pairing that varies over four decades (see inset of Fig.2), which corresponds to $|U|/t$ that varies from 1 to 8.

### The case of half-filled bipartite lattices

We consider the specific case of half-filled bipartite lattices where the number of orbitals in one sublattice is larger than that of the other, implying that at least one FB is located at $E = 0$. We propose to demonstrate the following remarkable property, valid for any $|U|$,

$$G_{\lambda\lambda}(\mathbf{r}) = \frac{1}{2}\delta(\mathbf{r}). \tag{5}$$

In a recent study [39] it has been shown that the Bogoliubov quasi-particle (QP) eigenstates present an interesting symmetry in half-filled systems. If $\mathcal{A}$ (resp. $\mathcal{B}$) denotes the first (resp. second) sublattice which contain $\Lambda_{\mathcal{A}}$ (resp. $\Lambda_{\mathcal{B}}$) orbitals per unit cell, the QP eigenstates can be subdivided in two families $\mathcal{S}^+$ and $\mathcal{S}^-$ defined in what follows. First, a generic QP eigenstate (in momentum space) has the form $|\Psi\rangle = (|\Psi^\uparrow\rangle, |\Psi^\downarrow\rangle)^t$ where the first $\Lambda_{\mathcal{A}}$ (resp. next $\Lambda_{\mathcal{B}}$) rows of $|\Psi^\sigma\rangle$ are the components on sublattice $\mathcal{A}$ (resp. $\mathcal{B}$). This eigenstate belongs to the subspace $\mathcal{S}^+$ (resp. $\mathcal{S}^-$) if $|\Psi^\downarrow\rangle = \hat{M}|\Psi^\uparrow\rangle$ (resp. $|\Psi^\downarrow\rangle = -\hat{M}|\Psi^\uparrow\rangle$) where the matrix $\hat{M} = \text{diag}(\hat{1}_{\Lambda_{\mathcal{A}}}, -\hat{1}_{\Lambda_{\mathcal{B}}})$. Additionally, for any finite $|U|$, it has been shown in Ref. [39] that the subset $\mathcal{S}^-$ (respectively $\mathcal{S}^+$) consists exactly in $\Lambda_{\mathcal{B}}$ (respectively $\Lambda_{\mathcal{A}}$) eigenstates of positive or zero energy and $\Lambda_{\mathcal{A}}$ (respectively $\Lambda_{\mathcal{B}}$) eigenstates of strictly negative energy.

Now, start with the definition $G_{\lambda\lambda}(\mathbf{r}) = \frac{1}{N_c}\sum_{\mathbf{k}} e^{i\mathbf{k}\cdot\mathbf{r}}\langle\hat{O}_{\lambda\mathbf{k},\uparrow}\rangle$ where , $\hat{O}_{\lambda\mathbf{k},\uparrow} = c^\dagger_{\mathbf{k}\lambda,\uparrow}c_{\mathbf{k}\lambda,\uparrow}$. At $T = 0$, its grand canonical average is given by,

$$\langle\hat{O}_{\lambda\mathbf{k},\uparrow}\rangle = \sum_m \langle\Psi^<_{m\mathbf{k}}|\hat{O}_{\lambda\mathbf{k},\uparrow}|\Psi^<_{m\mathbf{k}}\rangle, \tag{6}$$

where $|\Psi^<_{m\mathbf{k}}\rangle$ are the QP eigenstates of the BdG Hamiltonian of negative energy, $m$ being band index. Using the closure relation, $\sum_{m,s=<,>} |\Psi^s_{m\mathbf{k}}\rangle\langle\Psi^s_{m\mathbf{k}}| = 1$, where the sum runs over QP eigenstates with positive ($s =>$) and negative energy ($s =<$) and the symmetry mentioned above one can show that,

$$\sum_m \langle\Psi^<_{m\mathbf{k}}|\hat{O}_{\lambda\mathbf{k},\uparrow}|\Psi^<_{m\mathbf{k}}\rangle = \sum_m \langle\Psi^>_{m\mathbf{k}}|\hat{O}_{\lambda\mathbf{k},\uparrow}|\Psi^>_{m\mathbf{k}}\rangle, \tag{7}$$

which combined with Eq.6 leads to $\langle\hat{O}_{\lambda\mathbf{k},\uparrow}\rangle = \frac{1}{2}$ and demonstrates Eq.5.

It is interesting to remark that our proof can be straightforwardly extended to the case of disordered systems that preserve the bipartite character of the lattice, such as the presence of vacancies or bond disorder.

### The Stub lattice

The stub lattice is bipartite and offers the possibility to tune the QM without changing the of the nature of

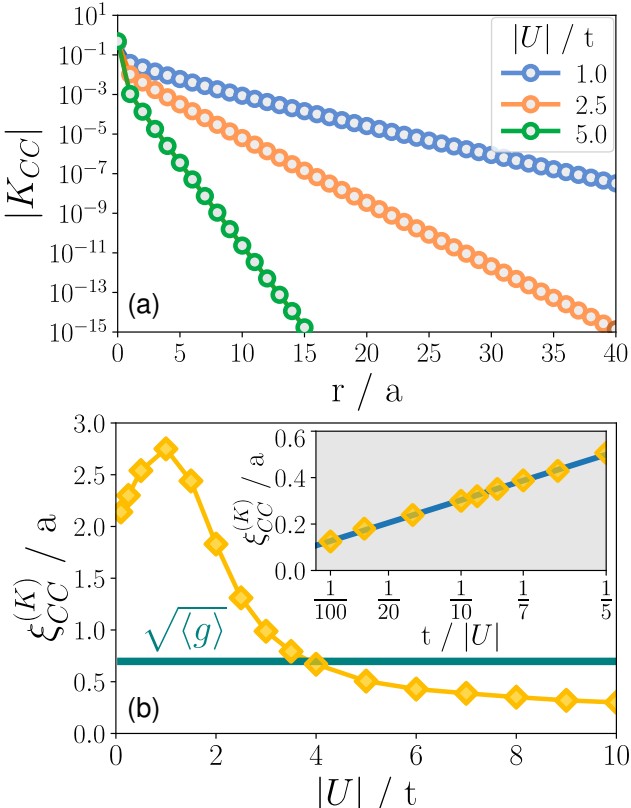

FIGURE 3. **(a)** $K_{CC}$ as a function of $r$ in the stub lattice for several values of $|U|/t$ (1, 2.5 and 5). **(b)** $\xi_{CC}^{(K)}$ as a function of $|U|$. The (dark-green) horizontal line depicts the square root of the quantum metric $\langle g \rangle$. The inset shows $\xi_{CC}^{(K)}$ for $|U| \gg t$. Here, $\alpha$ is set to 0.5 (see Fig.1) and the carrier density is fixed to $n = 3$ which corresponds to half-filling.

the compact localized eigenstates. The QM is controlled by the A-C hopping $(\alpha t)$(see Fig.1) and given by $\langle g \rangle = \frac{1}{2|\alpha|\sqrt{2+\alpha^2}}$ [40]. The stub lattice has been studied in great details in Refs. [18, 41]. Here, we restrict our study to the case $\alpha = 0.5$ and $n = 3$ which corresponds to a half-filled FB with $\langle g \rangle \simeq 0.7$.

First, one can already conclude from the previous section that the conventional CFs $(G_{\lambda\lambda})$ are given by Eq.5, which is indeed what we find numerically for any $|U|$ and any $\alpha$. Figure 3(a) depicts the anomalous CF $K_{CC}$ as a function of $|\mathbf{r}|$ for several values of $|U|$ which correspond to weak, intermediate and strong coupling regime. As it can be clearly seen, in all cases this CF decays exponentially with a lengthscale $\xi_{CC}^{(K)}$ (Cooper pair size) that reduces rapidly as $|U|$ increases. The variation of the extracted lengthscale $\xi_{CC}^{(K)}$ is plotted as a function of $|U|/t$ in Fig.3(b). In the limit of vanishing $|U|/t$ it is approximately (for this value of $\alpha$) $2a$, then it increases and reaches a maximum for $|U|/t = 1.5$ and beyond it decreases continuously. There is no simple explanation for the origin of this maximum, since for larger values

of $\alpha$ it disappears. The inset represents, its behaviour in the large $|U|/t$ limit. It is found that $\xi_{CC}^{(K)} \to 0.125\,a$. As it can be seen, $\xi_{CC}^{(K)}$ crosses $\sqrt{\langle g \rangle} = 0.7\,a$ at $|U|/t \approx 4$ and converges to a much smaller value. The large $|U|/t$ behaviour, is consistent with the fact that in the BEC regime, the Cooper pair size is expected to be very small. Remark that $K_{BB}$ and $K_{AA}$ vary similarly with the same lengthscale.

### The sawtooth chain

In contrast to the stub lattice, the sawtooth chain as illustrated in Fig.1(b), is a non bipartite lattice and does not allow the tuning of the QM. The FB exists only when the AB-hoppings (1st and 2nd neighbours) are $-\sqrt{2}t$. The superconductivity in the stub lattice has been addressed in details in Ref. [38] using a numerically exact method : the DMRG. It has been shown that the BdG approach reproduces accurately the exact results, for both the pairings and the superfluid weight. In Fig.4(a), both $G_{AA}$ and $K_{AA}$ are plotted as a function of $|\mathbf{r}|$, for different values of $|U|$. Here, the electron density is set to $n = 3$ which corresponds to the half-filled FB. As it can be seen, the lengthscales associated to the decay of $G_{AA}$ and $K_{AA}$ are almost identical both in the weak and strong coupling regime. Additionally, the slope appears to vary weakly. Notice that $G_{BB}$ and $K_{BB}$ behave similarly. Fig.4(b) depicts the variation of $\xi_{AA}^{(K)}$ as a function of $t/|U|$. The inset describes the weak coupling regime. In this regime, $\xi_{AA}^{(K)} \approx 0.735\,a$ and almost insensitive to $|U|$. As $|U|$ increases further, $\xi_{AA}^{(K)}$ decays monotonously. As seen in the case of the stub lattice, $\xi_{AA}^{(K)}$ crosses $\sqrt{\langle g \rangle}$ when $t/|U| \approx 0.05$ and converges towards $0.2\,a$. In the sawtooth chain, it can be shown that the minimal QM is $\langle g \rangle = \frac{1}{4\sqrt{3}}$. We should mention as well that our values of $G_{AA}(r)$ are consistent with the DMRG calculations of Ref. [38].

### The Creutz ladder

The Creutz ladder depicted in Fig1(c) is particularly interesting since its dispersion consists only in FBs, located at $E = \pm 2t$ in the non-interacting case. As a consequence of the uniform pairings, these bands remain flat when $|U|$ is non-zero. The superconductivity in the Creutz ladder have been addressed exactly, within the DMRG approach in Refs [12, 38]. As in the case of the sawtooth chain, it has been revealed that pairings and superfluid weight are accurately captured by the BdG theory. The A and B sites being equivalent, we focus our attention on $|K_{AA}|$ and $|G_{AA}|$. In addition, we consider the case of the quarter filled ladder (half-filled lower FB) which corresponds to $n = 1$. Both CFs are plotted in

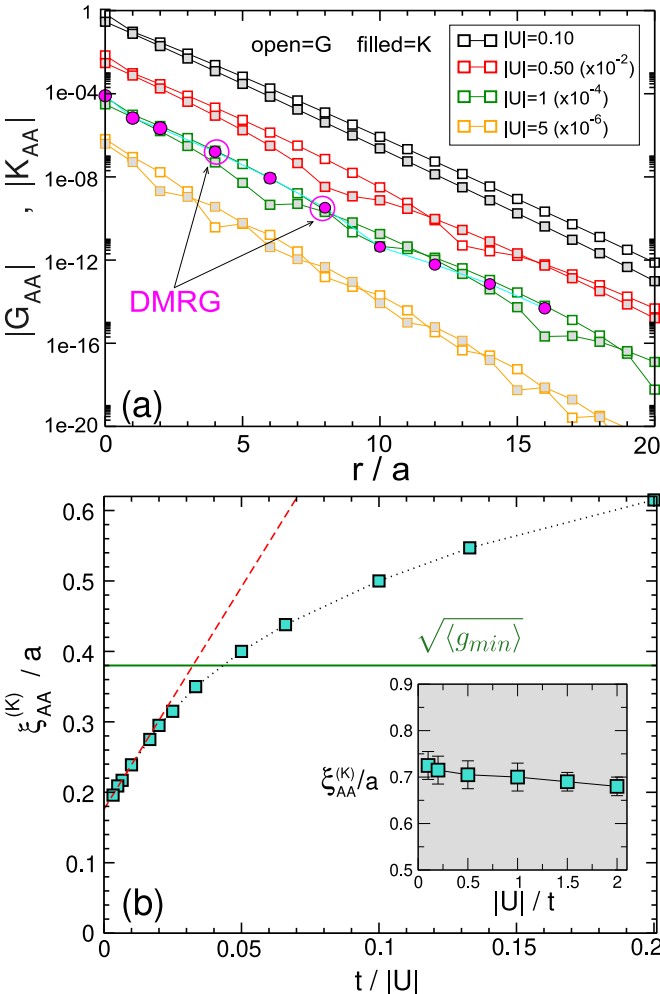

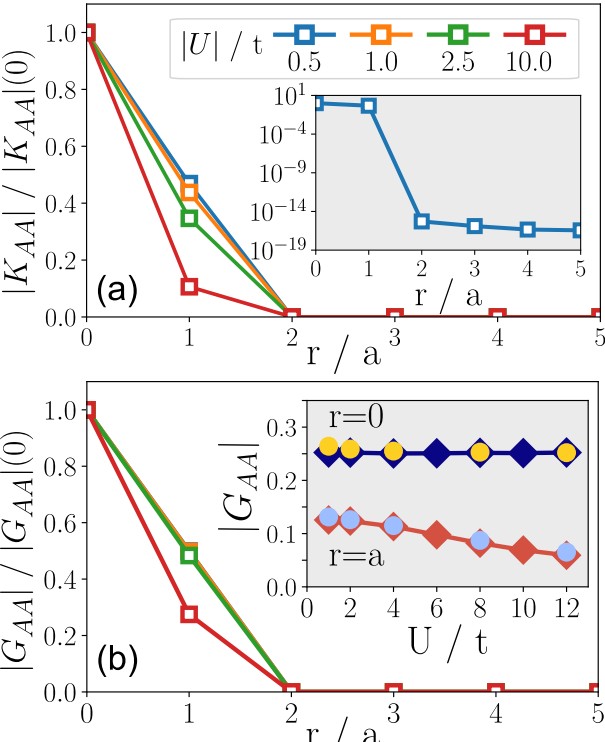

FIGURE 5. (a) $|K_{AA}|$ and (b) $|G_{AA}|$, rescaled by their value at $r = 0$, as a function of $r$ in the Creutz ladder for several values of $|U|$. The charge density is fixed $n = 1$. For $r \geq 2a$, both $|K_{AA}|$ and $|G_{AA}|$ are zero within our numerical precision. The inset in (a) represents $|K_{AA}|$ in log scale. The inset in (b) shows $|G_{AA}|$ as a function of $|U|$ for $r = 0$ and $r = a$. Diamonds are our calculations and circles are the DMRG data of Ref. [12]

FIGURE 4. (a) $|G_{AA}|$ and $|K_{AA}|$ as a function of $r$ in the sawtooth chain for several values of $|U|$. For the sake of clarity, $|G_{AA}|$ and $|K_{AA}|$ have been multiplied by $10^{-2}, 10^{-4}$ and $10^{-6}$ for $|U| = 0.5$, 1 and 5 respectively. The carrier density is $n = 3$ (half-filled FB). DMRG data for $|U| = 1$ from Ref. [38] are shown as well. (b) $\xi_{AA}^{(K)}$ as a function of $t/|U|$. The horizontal lines depicts the square root of the minimal quantum metric $\langle g_{min} \rangle$. The inset represents $\xi_{AA}^{(K)}$ as a function of $|U|$ for small values of $|U|$. The dashed red line is a linear fit for $\frac{t}{|U|} \leq 0.03$.

case of weak coupling that the CFs are given by,

$$G_{AA}(r) = K_{AA}(r) = \frac{1}{4}\delta_{r,0} - \frac{i}{8}\delta_{r,a} + \frac{i}{8}\delta_{r,-a},$$

$$G_{AB}(r) = K_{AB}(r) = \frac{1}{8}(\delta_{r,a} + \delta_{r,-a}). \tag{8}$$

We point out the fact that the analytic expression found for $G_{AA}(r)$ is consistent with the exact results obtained from DMRG calculations [12]. Indeed, it has been found (see Fig.10 in this manuscript) that for $r \geq 2a$, $G_{AA} \leq 10^{-12}$.

### The $\chi$-Lattice

The $\chi$-Lattice is a two dimensional system in which both electronic bands are dispersion-less and located at $E = \pm t$. As mentioned earlier this system has been introduced originally in Ref. [33]. The superconductivity has been addressed within the Quantum Monte Carlo method in Ref. [34] and within a mean field approach in Ref. [32]. We recall that the dimensionless parameter $\chi$

Fig.5 as a function of $|\mathbf{r}|$ for several values of $|U|$ ranging from weak to strong coupling regime. As it can be seen these two CFS behave similarly. Surprisingly, it is found that there are only two non-vanishing values corresponding respectively to $|\mathbf{r}| = 0$ and a. For larger distances, $|K_{AA}|$ and $|G_{AA}|$ are zero within the numerical accuracy. This is illustrated in the inset of Fig.5(b) where for $|\mathbf{r}| = 2a$ the CF $|G_{AA}|$ drops by 16 orders of magnitude. It is found as well that $|K_{AA}|(|\mathbf{r}| = a)$ decays very rapidly as $|U| \geq 1$ and eventually vanishes when $|U| \to \infty$. Thus, the Cooper pair size varies between 1 and 0 where 0 corresponds to $|U| = \infty$.

In the Appendix A, we demonstrate analytically in the

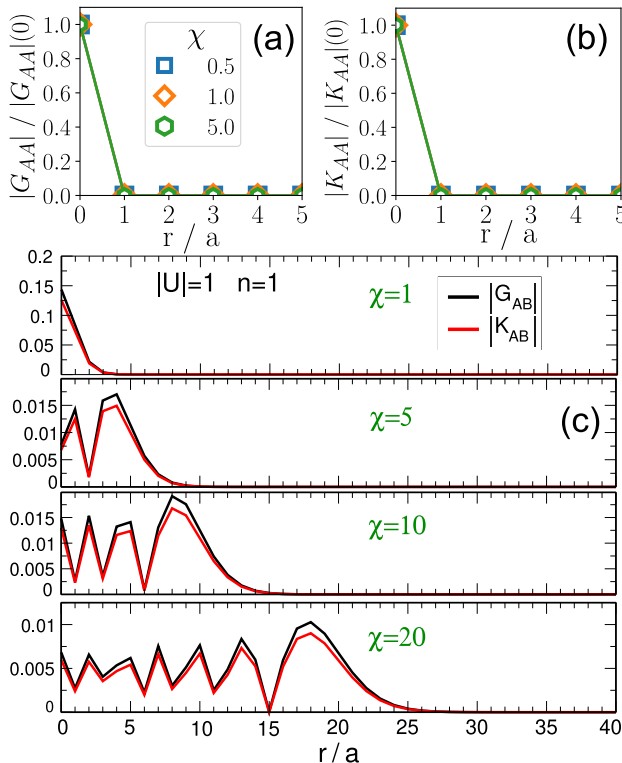

FIGURE 6. **(a)** and **(b)** $|G_{AA}|$ and $|K_{AA}|$ as a function of $r$ (along the $x$−direction) in the $\chi$-Lattice for several values of $\chi$. **(c)** same as in (a) and (b) for the off-diagonal correlation functions $|G_{AB}|$ and $|K_{AB}|$. The carrier density is $n = 1$ and the Hubbard parameter $|U| = 1$.

controls both the range of the hoppings and the value of the QM. Here, we focus on the quarter filled system which corresponds to a charge density $n = 1$. As it is the case in the Creutz ladder, the orbitals A and B are equivalent, pairings are identical on both sites. In addition, because the long range hoppings connects A to B sites only, this lattice is bipartite as well.

Let us now discuss our results. First, for any value of both $|U|$ and $\chi$, with high numerical accuracy we find,

$$\frac{4}{n}G_{\lambda\lambda}(\mathbf{r}) = \frac{|U|}{\Delta}K_{\lambda\lambda}(\mathbf{r}) = \delta(\mathbf{r}), \qquad (9)$$

where $\lambda = A, B$. These features are illustrated in Fig.6 (a) and (b). It should be emphasised that the property given in Eq.5 concerns only the case of half-filled bipartite lattices. Here, our system is quarter filled, which means that our findings are specific to the $\chi$-L. As a consequence, for any $|U|$ the Cooper pair size is zero. In the Appendix B, we have demonstrated analytically Eq.9 in the weak coupling regime.

More strikingly, we have found that the off-diagonal correlation functions $|G_{AB}|$ and $|K_{AB}|$ exhibit an unexpected behaviour as it can be clearly seen in Fig.6(c). First, one finds that $|G_{AB}|$ and $|K_{AB}|$ are very similar for any value of $\chi$. Furthermore, for a given $\chi$, one can

distinguish two distinct regimes. First, for $|\mathbf{r}| \leq \chi a$ the CFs oscillates as $|\mathbf{r}|$ increases. Secondly, when $|\mathbf{r}| \geq \chi a$ it decays monotonously as the distance increases. However, any attempt to fit the tail by a function of the form $r^{-b}e^{-|\mathbf{r}|/c}$ is unsuccessful. Hence, one cannot extract any characteristic lengthscale from these off-diagonal correlation functions. In the Appendix B, we have calculated analytically the $G_{AB}$ and $K_{AB}$ as a function of $\mathbf{r}$ in the limit of small values $|U|$. It is shown that $G_{AB}(\mathbf{r}) = K_{AB}(\mathbf{r}) = \frac{1}{4N_c}\sum_{\mathbf{k}}e^{i\mathbf{k}.\mathbf{r}}e^{-i\gamma_{\mathbf{k}}}$. This means that for this specific lattice the off-diagonal CFs coincides up to a coefficient with the (A,B) hoppings in real-space. In addition, in the limit of large $|\mathbf{r}|$ along the $x$-direction, it is shown that,

$$G_{AB}(\mathbf{r}) \propto (-i)^{n_x}J_0(\chi)\frac{1}{\sqrt{2\pi n_x}}e^{n_x.ln(\frac{e\chi}{2n_x})}, \qquad (10)$$

where $\mathbf{r} = (n_x a, 0)$ and $J_0$ is the Bessel function of the first kind and order 0. This clarifies why we could not extract a typical lengthscale from the numerical data plotted in Fig.6(c).

**Connection with recent studies**

In this paragraph, we would like to discuss the connection between our findings and recent studies [31, 32]. It is claimed in these articles, that the coherence length in quasi FBs can be expressed as, $\xi_c = \sqrt{\xi_{BCS}^2 + \langle g \rangle}$ where $\langle g \rangle$ is the average of the quantum metric (minimal). The BCS contribution vanishes when the band is rigorously flat. In Ref. [32], the authors have illustrated their point by considering the $\chi$−Lattice. Based on the fact that the coherence length is extracted from the long distance decay of $K_{\lambda\lambda}$ our findings clearly contradicts their prediction. Indeed, in the specific case of the $\chi$−Lattice we have found that for any $|U|$ and any $\chi$, thus any QM, the size of the Cooper pair is always zero. What is the origin of this contradiction? In Ref. [32], the authors have treated the electron-electron correlation at the mean field level, similar to ours. However, their decoupling of the Hubbard term is performed in momentum space instead of real space as done within the BdG approach. Additionally, the authors have performed a projection of the fermionic operators onto the lowest FB which is, in their work, the key step so that the QM can emerge. In contrast, in our BdG approach we perform no projection and treat essentially the electron-electron correlation at the same level, thus it is striking that our results differ from those of Ref. [32]. To extract the coherence length, the authors consider the pair-pair correlation function of the form, $C_{\lambda\eta}(\mathbf{r}) = \langle\hat{c}_{i\lambda\uparrow}^{\dagger}\hat{c}_{i\eta\downarrow}^{\dagger}\hat{c}_{0\eta\downarrow}\hat{c}_{0\lambda\uparrow}\rangle$, which within mean field theory leads to,

$$C_{\lambda\eta}(\mathbf{r}) \xrightarrow{MF} G_{\lambda\lambda}(\mathbf{r})G_{\eta\eta}(\mathbf{r}) + |K_{\lambda\eta}(\mathbf{0})|^2. \qquad (11)$$

This result clearly differs from that found in Ref. [32] and highlight the absence of the emergence of the quantum metric. Furthermore, as shown in Appendix C, projection onto flat bands is not in question ; the real problem seems to be an error in the calculation procedure (decoupling of correlation functions).

## CONCLUSION

To conclude, we have investigated the normal and anomalous correlations functions in various flat band systems and extracted the associated characteristic lengthscales. It is found in this study that the size of the Cooper pairs is comparable to the lattice spacing, both in the weak and strong coupling regime. Independently of how extended the hoppings are, it is revealed as well that the normal correlation functions reduce to a Dirac function in the case of half-filled bipartite lattices. In contrast with a recent claim, our findings indicate that the coherence length as extracted from the decay of the anomalous correlation function appears to be disconnected from the quantum metric ($\sqrt{\langle g \rangle}$). We have provided some arguments that may clarify the origin of the disagreement between our results and these recent studies. It is interesting to remark that besides the disagreement quoted above, the values of the coherence length found for the sawtooth chain and for the Stub lattice are comparable to $\sqrt{\langle g \rangle}$. Finally, we believe that this study could as well motivate new reflections on the concept of coherence length in flat band systems and perhaps on alternative ways of defining this characteristic lengthscale of the Cooper pairs from a theoretical point of view.

## AKNOWLEDGEMENT

We thank George Batrouni, Si Min Chan and Benoit Grémaud for kindly sending us their DMRG data.

## APPENDIX A : THE CORRELATION FUNCTIONS IN THE CREUTZ LADDER

In this appendix we propose to derive analytically the correlations functions G and K as defined in the main text in the quarter filled Creutz ladder. We focus our attention on small values of $|U|$. We restrict our calculation to $T = 0$. The BdG Hamiltonian reads,

$$\hat{H}_{BdG} = \sum_k \hat{\Psi}_k^\dagger \begin{pmatrix} \hat{h}_{\mathbf{k}}^\uparrow & \Delta \hat{1}_{2\times 2} \\ \Delta^* \hat{1}_{2\times 2} & -\hat{h}_{-\mathbf{k}}^{\downarrow *} \end{pmatrix} \hat{\Psi}_k, \quad \text{(A.1)}$$

where we have introduced the Nambu spinor $\hat{\Psi}_k^\dagger = (\hat{c}_{Ak\uparrow}, \hat{c}_{Bk\uparrow}, \hat{c}_{A-k\downarrow}, \hat{c}_{B-k\downarrow})^t$ and the block matrix,

$$\hat{h}_{\mathbf{k}}^\uparrow = \begin{pmatrix} -2t\sin(ka) - \tilde{\mu} & -2t\cos(ka) \\ -2t\cos(ka) & 2t\sin(ka) - \tilde{\mu} \end{pmatrix}, \quad \text{(A.2)}$$

where we have introduced $\tilde{\mu} = \mu + \frac{|U|}{4}n$. Because of time reversal symmetry $\hat{h}_{-\mathbf{k}}^{\downarrow *} = \hat{h}_{\mathbf{k}}^\uparrow$. Notice as well that the pairing $\Delta$, uniform because A and B sites are equivalent, can be taken real. Here, the total carrier density $n = 2n_A = 2n_B$ is set to 1.

First, we consider the case $|U| = 0$ for which the chemical potential $\mu = \mu_0 = -2t$. The quasi-particle (QP) eigenvalues are $E_{1,4} = \pm 4t$, and $E_{2,3} = 0$ which is doubly degenerate. The corresponding QP eigenstates are of the form, $|\Psi_i\rangle = (|\psi_i^\uparrow\rangle, |\psi_i^\downarrow\rangle)^t$, where $i = 1, .., 4$. More precisely they are given by, $|\Psi_1^0\rangle = (0, |\phi_0^+\rangle)^t$, $|\Psi_2^0\rangle = (|\phi_0^-\rangle, 0)^t$, $|\Psi_3^0\rangle = (0, |\phi_0^-\rangle)^t$, and $|\Psi_4^0\rangle = (|\phi_0^+\rangle, 0)^t$, where,

$$|\phi_0^\pm\rangle = \frac{1}{\sqrt{2}} \frac{1}{\sqrt{1 \pm \sin(ka)}} \begin{pmatrix} -\cos(ka) \\ \sin(ka) \pm 1 \end{pmatrix}. \quad \text{(A.3)}$$

When the Hubbard term is switched on, we apply a pertubation theory for degenerate pair eigenstates ($|\Psi_2^0\rangle, |\Psi_3^0\rangle$) that leads to, $E_\pm = \pm\sqrt{(\delta\tilde{\mu})^2 + \Delta^2}$ where $\delta\tilde{\mu} = \tilde{\mu} - \mu_0$. The corresponding QP eigenstates are,

$$|\Psi_\pm\rangle = \frac{1}{\sqrt{N^\pm}} \left( \Delta|\Psi_2^0\rangle + (\delta\tilde{\mu} \pm \sqrt{(\delta\tilde{\mu})^2 + \Delta^2})|\Psi_3^0\rangle \right), \text{(A.4)}$$

where $N^\pm = 2\left( \delta\tilde{\mu}^2 + \Delta^2 \pm \delta\tilde{\mu}\sqrt{(\delta\tilde{\mu})^2 + \Delta^2} \right)$.

Using the self-consistent equations for the carrier density which for each spin sector is $1/4$ and the gap equation one finds in the limit of small $|U|$,

$$\delta\tilde{\mu} = 0 + o(|U|^2), \quad \text{(A.5)}$$

$$\Delta = \frac{|U|}{4} + o(|U|^2). \quad \text{(A.6)}$$

Thus, the QP eigenstates take the simple form $|\Psi_\pm\rangle = \frac{1}{\sqrt{2}}(|\Psi_2^0\rangle \pm |\Psi_3^0\rangle)$, their respective energy being $E_\pm = \mp\frac{1}{4}|U|$.

Using the expressions of $|\phi_0^\pm\rangle$ as given in Eq.A.3, one finds, $\langle c_{Ak,\uparrow}^\dagger c_{Ak,\uparrow}\rangle = \langle c_{Ak,\uparrow}^\dagger c_{Ak,\downarrow}\rangle = \frac{1}{4}(1 + \sin(k))$. After a trivial Fourier transform, we finally end up with,

$$G_{AA}(r) = K_{AA}(r) = \frac{1}{4}\delta_{r,0} - \frac{i}{8}\delta_{r,a} + \frac{i}{8}\delta_{r,-a}. \quad \text{(A.7)}$$

In addition for the off-diagonal CFs it is found that,

$$G_{AB}(r) = K_{AB}(r) = \frac{1}{8}(\delta_{r,a} + \delta_{r,-a}). \quad \text{(A.8)}$$

These results explain the data plotted in Fig. 5 of the present manuscript. We recall that our proof is restricted to $|U| \leq t$.

## APPENDIX B : THE CORRELATION FUNCTIONS IN THE $\chi$-LATTICE

In this appendix, our purpose is to derive analytically the correlation functions G and K in the quarter filled $\chi$−Lattice. The BdG calculations are performed for small values of the Hubbard parameter $|U|$ at $T = 0\ K$. The BdG Hamiltonian has the same form as that given in Eq.A.1 of the Appendix A, with $\hat{h}_{\mathbf{k}}^{\uparrow}$ now given by,

$$\hat{h}_{\mathbf{k}}^{\uparrow} = \begin{pmatrix} -\mu - \frac{|U|}{4}n & -te^{-i\gamma_{\mathbf{k}}} \\ -te^{i\gamma_{\mathbf{k}}} & -\mu - \frac{|U|}{4}n \end{pmatrix}, \quad \text{(B.1)}$$

where $\gamma_{\mathbf{k}} = \chi(\cos(k_x a) + \cos(k_y a))$.
Notice that the $\chi$−Lattice is both bipartite and time reversal symmetric as well which implies $\hat{h}_{-\mathbf{k}}^{\downarrow *} = \hat{h}_{\mathbf{k}}^{\uparrow}$.
To calculate the QP eigenstates, we use the same notation as those of Appendix A. At $|U| = 0$, the quasi-particle (QP) eigenstates are located at $E_{1,4} = \pm 2\,t$, and $E_{2,3} = 0$ which is doubly degenerate, the chemical potential being $\mu = \mu_0 = -\,t$. The one particle eigenstates read,

$$|\phi_0^{\pm}\rangle = \frac{1}{\sqrt{2}} \begin{pmatrix} \mp e^{-i\frac{\gamma_{\mathbf{k}}}{2}} \\ e^{i\frac{\gamma_{\mathbf{k}}}{2}} \end{pmatrix}. \quad \text{(B.2)}$$

The equations (A.4), (A.5) and (A.6) of Appendix A are valid as well in the case of the $\chi$-Lattice at quarter filling. Thus one straightforwardly gets, $\langle \hat{c}_{A\mathbf{k},\uparrow}^{\dagger} \hat{c}_{A\mathbf{k},\uparrow} \rangle = \frac{1}{4}$, $\langle \hat{c}_{A\mathbf{k},\uparrow}^{\dagger} \hat{c}_{B\mathbf{k},\uparrow} \rangle = \frac{1}{4}e^{-i\gamma_{\mathbf{k}}}$, $\langle \hat{c}_{A\mathbf{k},\uparrow}^{\dagger} \hat{c}_{A-\mathbf{k},\downarrow}^{\dagger} \rangle = \frac{1}{4}$ and $\langle \hat{c}_{A\mathbf{k},\uparrow}^{\dagger} \hat{c}_{B-\mathbf{k},\downarrow}^{\dagger} \rangle = \frac{1}{4}e^{-i\gamma_{\mathbf{k}}}$. It follows that,

$$G_{AA}(\mathbf{r}) = K_{AA}(\mathbf{r}) = \frac{1}{4}\delta_{\mathbf{r},\mathbf{0}}, \quad \text{(B.3)}$$

and the off-diagonal CFs are,

$$G_{AB}(\mathbf{r}) = K_{AB}(\mathbf{r}) = \frac{1}{4}f_{AB}(\mathbf{r}), \quad \text{(B.4)}$$

where, we have introduced $f_{AB}(\mathbf{r}) = \frac{1}{N_c}\sum_{\mathbf{k}}e^{i\mathbf{k}\cdot\mathbf{r}}e^{-i\gamma_{\mathbf{k}}}$. Thus, $G_{AB}(\mathbf{r})$ and $K_{AB}(\mathbf{r})$ coincide, up to a constant, with the (A,B) hoppings. We now propose to calculate the analytic expression of $f_{AB}(\mathbf{r})$ for both $|\mathbf{r}|/a \leq \chi$ and $|\mathbf{r}|/a \gg \chi$.
Let us write $\mathbf{r} = (n_x, n_y)$, $f_{AB}(\mathbf{r})$ can be rewritten as the following product,

$$f_{AB}(\mathbf{r}) = I_{n_x}(-i\chi) \cdot I_{n_y}(-i\chi), \quad \text{(B.5)}$$

where $I_n(i\chi) = \frac{1}{2\pi}\int_{-\pi}^{+\pi}e^{in\theta}e^{i\chi\cos(\theta)}$ is the modified Bessel function of the first kind and order $n$. We can now rely on the properties of the Bessel functions such as $I_n(-i\chi) = (-i)^n J_n(\chi)$ which leads to,

$$f_{AB}(\mathbf{r}) = (-i)^{n_x+n_y}J_{n_x}(\chi) \cdot J_{n_y}(\chi). \quad \text{(B.6)}$$

In the regime where $|\mathbf{r}| \leq \chi a$ one can expand the Bessel function [42],

$$J_n(\chi) \simeq \sqrt{\frac{2}{\pi\chi}}\cos\left(\chi - n\frac{\pi}{2} - \frac{\pi}{4}\right), \quad \text{(B.7)}$$

and similarly for $J_m(\chi)$. This clearly explains the presence of the oscillations observed in Fig. 6 of the manuscript.
In the opposite limit, more precisely for $\chi \ll \sqrt{|n_x| + |n_y|}$, one has,

$$J_n(\chi) \simeq \frac{1}{\Gamma(n+1)}\left(\frac{\chi}{2}\right)^n. \quad \text{(B.8)}$$

According to the well known Stirling formula, for $n \gg 1$ one can write $\Gamma(n+1) \simeq \frac{1}{\sqrt{2\pi n}}(\frac{n}{e})^n$. Thus, along the $x$−direction for instance, it implies the following result,

$$f_{AB}(\mathbf{r}) = (-i)^{n_x}\frac{J_0(\chi)}{\sqrt{2\pi n_x}}e^{n_x\ln\left(\frac{e\chi}{2n_x}\right)}. \quad \text{(B.9)}$$

This equation explains (i) the rapid decay observed in Fig. 6 of our manuscript and (ii) the impossibility to extract a characteristic lengthscale from the decay at large distance of the off-diagonal correlation functions.

## APPENDIX C : POSSIBLE ORIGIN OF THE DISAGREEMENT WITH REF. [32]

In this appendix our goal is to point out the possible origin of the disagreement between our findings and those of Ref. [32]. In this study the coherence length is extracted from the exponential decay of,

$$\mathcal{C}(\mathbf{r}_i, \mathbf{r}_j) = \sum_{\alpha\beta}\langle c_{i\alpha,\uparrow}c_{i\beta,\downarrow}c_{j\beta,\downarrow}^{\dagger}c_{j\alpha,\uparrow}^{\dagger}\rangle. \quad \text{(C.1)}$$

From Eq. (C.1), the authors have perform the projection of the fermionic operators ($c_{\mathbf{k}\lambda,\sigma}^{\dagger}$) onto the the flat band operators ($\bar{c}_{\mathbf{k},\sigma}^{\dagger}$) and then evaluate the expectation value $\langle\ldots\rangle$ with Wick's theorem. The projection reads $c_{\mathbf{k}\lambda,\sigma}^{\dagger} \to u_{\lambda\mathbf{k},\sigma}\bar{c}_{\mathbf{k},\sigma}^{\dagger}$, where $u_{\lambda\mathbf{k},\sigma}$ is the flat band eigenstate. Then, they define $\Lambda_{\mathbf{k},p} = \sum_{\lambda}u_{\lambda\mathbf{k},\uparrow}u_{\lambda-\mathbf{p},\downarrow}$ and use the time-reversal symmetry that reads : $u_{\lambda-\mathbf{p},\downarrow} = u_{\lambda\mathbf{p},\uparrow}^* = u_{\lambda\mathbf{p}}^*$. The authors find,

$$\mathcal{C}(\mathbf{r}_i, \mathbf{r}_j) = \frac{1}{N_c}\overbrace{\sum_{\mathbf{k}\mathbf{k}'}\langle\bar{c}_{\mathbf{k},\uparrow}\bar{c}_{-\mathbf{k},\downarrow}\rangle\langle\bar{c}_{-\mathbf{k}',\downarrow}^{\dagger}\bar{c}_{\mathbf{k}',\uparrow}^{\dagger}\rangle}^{\mathbf{r}-independent} \quad \text{(C.2)}$$

$$+ \frac{1}{N_c}\sum_{\mathbf{k}\mathbf{q}}e^{-i\mathbf{q}\cdot(\mathbf{r}_i-\mathbf{r}_j)}|\Lambda_{\mathbf{k}+\mathbf{q},\mathbf{k}}|^2\langle\bar{c}_{\mathbf{k}+\mathbf{q},\uparrow}\bar{c}_{\mathbf{k}+\mathbf{q},\uparrow}^{\dagger}\rangle\langle\bar{c}_{-\mathbf{k},\downarrow}\bar{c}_{-\mathbf{k},\downarrow}^{\dagger}\rangle.$$

The key quantity is the second term which contains $|\Lambda_{\mathbf{k}+\mathbf{q},\mathbf{k}}|^2$. A quadratic expansion in the limit $\mathbf{q} \to 0$ yields $|\Lambda_{\mathbf{k}+\mathbf{q},\mathbf{k}}|^2 \overset{|\mathbf{q}|\to 0}{=} 1 - \sum_{\mu\nu}g_{\mu\nu}(\mathbf{k})q^{\mu}q^{\nu} + o(|\mathbf{q}|^2)$, where $g_{\mu\nu}(\mathbf{k})$ is the quantum metric.
We disagree with Eq. (C.2). Starting from Eq. (C.1) and

after the projection on the FB operators, we get instead,

$$\mathcal{C}(\mathbf{r}_i, \mathbf{r}_j) = \frac{1}{N_c} \sum_{\mathbf{k}\mathbf{k}'} |\Lambda_{\mathbf{k},\mathbf{k}'}|^2 \langle \bar{c}_{\mathbf{k}',\uparrow} \bar{c}_{-\mathbf{k}',\downarrow}\rangle \langle \bar{c}^\dagger_{-\mathbf{k},\downarrow} \bar{c}^\dagger_{\mathbf{k},\uparrow}\rangle$$
$$+ \frac{1}{N_c} \sum_{\mathbf{k}\mathbf{q}} e^{-i\mathbf{q}\cdot(\mathbf{r}_i - \mathbf{r}_j)} \langle \bar{c}_{\mathbf{k}+\mathbf{q},\uparrow} \bar{c}^\dagger_{\mathbf{k}+\mathbf{q},\downarrow}\rangle \langle \bar{c}_{-\mathbf{k},\downarrow} \bar{c}^\dagger_{-\mathbf{k},\uparrow}\rangle. \tag{C.3}$$

The first term is a constant ($\mathbf{r}$-independent) while the second one does not depend on $\Lambda_{\mathbf{k}+\mathbf{q},\mathbf{k}}$. Thus, in contrast to their calculation, the quantum metric does not emerge in the correlation function. Furthermore, the projection onto the flat bands is not in question.

————

\* maxime.thumin@neel.cnrs.fr
† georges.bouzerar@neel.cnrs.fr

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

# Referee 1

**Referee**:
1- I encourage the authors to deepen their discussion of its connection with recent studies. The authors relate the deviation of their results to those within the literature to the decoupling of the Hubbard term and the projection of the operators onto the lowest flatband. Throughout the manuscript, the authors have already justified their approach by relating it to DMRG calculations. It would strengthen their conclusions when the authors:(i) Be more mathematically precise about how the decoupling differs,
(ii) Discuss how the projection mechanism may account for the different findings, and
(iii) Give arguments, which approach is expected to capture the experimental results, and give, if possible, corresponding references.

 I understand that a complete answer is far beyond the scope of this first work so I would be satisfied by a more in-depth description of the problem, related deviations seen in other literature, and a more in-depth guidance for future research.

**Answer:** We agree on the fact that a complete answer goes beyond the scope of our work.This is a key question since in our case and in the manuscript written by Hu, Chen and Law. (Ref.32 in our manuscript), the electronic correlations are treated the same way, thus one expects similar and even identical results. In order to find the possible origin of the disagreement between their calculation and our findings, we have redone the calculation given in Eq.(S15) of the third version of their manuscript *arXiv:2308.05686*. In this paper it is written in their Appendix,

$$\mathcal{C}(\boldsymbol{r},\boldsymbol{r}') = \sum_{\alpha\beta} \langle a_{\alpha\uparrow}(\boldsymbol{r}) a_{\beta\downarrow}(\boldsymbol{r}) a^{\dagger}_{\beta\downarrow}(\boldsymbol{r}') a^{\dagger}_{\alpha\uparrow}(\boldsymbol{r}') \rangle \qquad \text{(L1)}$$

$$= \frac{1}{N} \sum_{\boldsymbol{k}\boldsymbol{k}'\boldsymbol{q}} \sum_{\alpha\beta} e^{-i\boldsymbol{q}\cdot(\boldsymbol{r}-\boldsymbol{r}')} \langle a_{\alpha\uparrow}(\boldsymbol{k}'+\boldsymbol{q}) a_{\beta\downarrow}(-\boldsymbol{k}') a^{\dagger}_{\beta\downarrow}(-\boldsymbol{k}) a^{\dagger}_{\alpha\uparrow}(\boldsymbol{k}+\boldsymbol{q}) \rangle \qquad \text{(L2)}$$

$$\rightarrow \frac{1}{N} \sum_{\boldsymbol{k}\boldsymbol{k}'\boldsymbol{q}} e^{-i\boldsymbol{q}\cdot(\boldsymbol{r}-\boldsymbol{r}')} \Lambda(\boldsymbol{k}+\boldsymbol{q},\boldsymbol{k}) \dot{\Lambda}^{*}(\boldsymbol{k}'+\boldsymbol{q},\boldsymbol{k}') \langle c_{\uparrow}(\boldsymbol{k}'+\boldsymbol{q}) c_{\downarrow}(-\boldsymbol{k}') c^{\dagger}_{\downarrow}(-\boldsymbol{k}) c^{\dagger}_{\uparrow}(\boldsymbol{k}+\boldsymbol{q}) \rangle \qquad \text{(L3)}$$

$$= \frac{T}{N} \sum_{\boldsymbol{q}} e^{-i\boldsymbol{q}\cdot(\boldsymbol{r}-\boldsymbol{r}')} \sum_{n\boldsymbol{k}} |\Lambda(\boldsymbol{k}+\boldsymbol{q},\boldsymbol{k})|^{2} \mathrm{Tr} G(i\omega_n,\boldsymbol{k}) G(-i\omega_n,-\boldsymbol{k}+\boldsymbol{q}), \qquad \text{(L4)} \qquad \text{(S15)}$$

If we start from the first line (L1), we will correctly find the second one, Eq. (L2). However after the introduction of the projection operator, we do not get the equation (L3), but instead we find the following result,

$$\mathcal{C}(r,r') = \sum_{\mathbf{k}\mathbf{k}'\mathbf{q}} e^{-i\mathbf{q}\cdot(\mathbf{r}-\mathbf{r}')} \Lambda(\mathbf{k}+\mathbf{q},\mathbf{k}'+\mathbf{q}) \Lambda(\mathbf{k},\mathbf{k}')^{*} \langle \hat{c}_{\uparrow}(\mathbf{k}'+\mathbf{q}) \hat{c}_{\downarrow}(-\mathbf{k}') \hat{c}^{\dagger}_{\downarrow}(-\mathbf{k}) \hat{c}^{\dagger}_{\uparrow}(\mathbf{k}+\mathbf{q}) \rangle \qquad \text{(L3')}$$

Even a relabelling of the momentum variables in (L3') will not lead to Eq. (L3). On the other hand the mean field decoupling of Eq.(L3') leads to,

$$\mathcal{C}(r, r') = \sum_{\mathbf{k}\mathbf{k}'} |\Lambda(\mathbf{k}, \mathbf{k}')|^2 \langle \hat{c}_\uparrow(\mathbf{k}') \hat{c}_\downarrow(-\mathbf{k}') \rangle \langle \hat{c}_\downarrow^\dagger(-\mathbf{k}) \hat{c}_\uparrow^\dagger(\mathbf{k}) \rangle$$
$$+ \sum_{\mathbf{k}\mathbf{q}} e^{-i\mathbf{q}\cdot(\mathbf{r}-\mathbf{r}')} \langle \hat{c}_\uparrow(\mathbf{k}+\mathbf{q}) \hat{c}_\uparrow^\dagger(\mathbf{k}+\mathbf{q}) \rangle \langle \hat{c}_\downarrow(-\mathbf{k}) \hat{c}_\downarrow^\dagger(-\mathbf{k}) \rangle \qquad \text{(L4')}$$

This expression shows that the first term is a constant (**r**-independent) while the second one does not depend on $\Lambda(\mathbf{k+q,k})$. Thus, in contrast to their calculation, the quantum metric will not emerge in the correlation function. We can conclude that the origin of the problem and the disagreement with our findings is a mistake in their calculation.

We have added a third appendix (Appendix C) to clarify the origin of the disagreement between our results and those of Hu, CHen and Law.

**Referee**:
2- Some parts need some more explanations and context. In particular,
(i) What do the authors precisely mean by "intraband" and "interband nature" in the introduction?

**Answer:** In the case of the conventional superconductivity, which corresponds to single band physics, the superconducting order parameter and the superfluid weight are controlled by matrix elements of the current operator within the same band and the inter-band elements are negligible. In this case, the non zero Fermi velocity is crucial. In flat bands the fermi velocity vanishes, thus the intraband terms are zero, however the matrix elements between different QP bands are non zero and control the superfluid weight. We have added a sentence in the introduction regarding the explanation of the term 'interband' nature.

**Referee**:
(ii) Why should we have doubts about the previous literature on the relation between the quantum metric and coherence length in the first place?

**Answer:** This is related to the discussion , response to point 1. We asked ourselves whether the metric (which is a measure of a characteristic length squared) could emerge from the calculation of the coherence length as defined in our manuscript (ie from the decay of the correlations functions of the form $\langle c_\uparrow^\dagger(0) c_\downarrow^\dagger(r) \rangle$. Then, we discovered the results of Law et al. (arxiv paper 2023) that caught our attention. So we have decided to check their expressions and found inconsistent results. As shown in the discussion above we can now provide an explanation to the origin of the disagreement. We believe as well that our work and that of Law et al., should motivate further studies and push us to ask ourselves : "what is the correct definition of the coherence length".

**Referee**:
iii) Why are these four models chosen in light of potential deviations? What can the insights of each model contribute to the big picture?

**Answer:** We have chosen these 4 different models because they cover a broad family of flat band systems. As said in the manuscript this choice "*allows to estimate the impact of (i) the bipartite character of the lattice, (ii) the tunability of the quantum metric, (iii) the absence of dispersive bands in the spectrum, (iv) and last the impact of the lattice dimension.*"

**Referee**:
(iv) What is the role of the BCS - BEC crossover in light of the investigated deviation between existing and this literature?

**Answer:**
We are not sure we fully understand the question but we try to answer the question. To begin with, we wanted to show in a clear way that the procedure used to extract the coherence length is the correct one. Indeed, in the case of conventional superconductivity (Fermi energy in the dispersive band), we recover the expected BCS result very accurately. Furthermore, without any fitting parameter we find that our numerical data coincide with,

$$\xi = \frac{\hbar v_F}{\Delta}$$

Surprisingly it is found that this expression, expected to be valid only in the weak coupling regime (BCS regime), reproduces as well accurately the numerical data in the BEC region (strong coupling regime).

**Referee**:
3- The presentation requires some minor revisions:
(i) I would ask the authors to reduce the number of abbreviations to a minimum as they are making the document less accessible in particular for those not experienced with the particular subfield.

**Answer:** We thought that the use of abbreviations would not confuse the reader. However, following the referee's recommendation, we have removed in the modified version of the manuscript the abbreviation "QMC" , "SC", "SaL", "CrL", "StL" and "$\chi$-L".

**Referee**:
(ii) I encourage the authors to work on the presentation of their figures. Whereas several

of them are very clear and easily accessible such as Fig. 1 and 2, others lack a consistent choice of labels, fonts, size, etc.

**Answer:** We did not completely understand what should be changed in the figures, especially because we have spent a considerable amount of time in generating them. However, we have modified y-labels in Fig.5a and Fig.6a: $|U| |K_{AA}| / \Delta_A$ becomes $|K_{AA}| / |K_{AA}|(0)$, and $2 |G_{AA}| / n_A$ becomes $|G_{AA}| / |G_{AA}|(0)$.

**Referee**:
(iii) The authors might check for some typos/double-use of labels. An example of this would be the double use of the index alpha for both the orbital label and the hopping strength.

**Answer:**
We have corrected and checked the typos and double-use of labels. All the confusions regarding $\alpha$ have been corrected.

# Referee 2

**Referee**:
1- This work is overall well written and the numerical data and the derivations of the analytical results seem sound in my opinion. On the other hand, there are some clear flaws in the way the results are interpreted leading to the conclusion by the authors that the relation between coherence length and quantum metric is not valid. While I cannot express myself on the validity of this relation, the arguments proposed by the authors are in my opinion not solid enough to conclude that the coherence length is disconnected from the quantum metric.

**Answer:**
First, we would like to thank the referee for pointing out that "*This work is overall well written and the numerical data and the derivations of the analytical results seem sound in my opinion.*" Then the referee says: "*While I cannot express myself on the validity of this relation, the arguments proposed by the authors are in my opinion not solid enough to conclude that the coherence length is disconnected from the quantum metric.*". We have been fully transparent in the procedure used to calculate the coherence length. We have defined the anomalous Green's function:

$$K_{\alpha\beta}(r) = \langle c_{i\alpha\uparrow}\, c_{i+r\beta\downarrow}\rangle$$

that is used, in the conventional case, to extract the coherence length. As a test of validity, we have perfectly recovered the expected expression in the BCS limit and found that it is valid as well in the BEC regime:

$$\xi = \frac{\hbar v_F}{\Delta}$$

The agreement between the coherence length extracted from the exponential decay of the numerically calculated correlation function and the analytic expression is excellent for a pairing amplitude that varies over several decades (Fig.2 in the manuscript)! Based on this result, it is natural to use the function $K_{\alpha\beta}$ to extract the coherence length which is what is done in the present work. *The referee says: "On the other hand, there are some clear flaws in the way the results are interpreted leading to the conclusion by the authors that the relation between coherence length and quantum metric is not valid".* We have realized the calculations in the weak, intermediate and strong coupling regime and

as well considered different types of lattices and geometries. Our numerical results establish unambiguously that the coherence length, as it is extracted from the decay of the correlation function $K_{\alpha\beta}$ is disconnected from the quantum metric. Furthermore, to demonstrate the validity of our calculations, we also compared them with the exact results (found in the literature) and obtained using the DMRG approach. Thus we disagree with the referee's sentence.

Furthermore, we emphasize that we have now added in the manuscript a new appendix (appendix C) that clarifies the possible origin of the discrepancy between our findings and those of Law et al. The projection onto the flat band is not in question but there is a mistake in their calculations.

**Referee**:
2 - The first point is that the authors should clearly state the limits of validity of the relation between quantum metric and coherence length.

**Answer:**
This remark is connected to point 1. As said before we have performed our calculation for different lattices and geometries, in the weak, intermediate and strong coupling regime and our conclusion was systematically the same: there is no direct connection between the extracted length scales and the quantum metric.

**Referee**:
3- Indeed, I do not expect the relation to hold for arbitrary values of the interaction strength. **The benchmark in this sense is the well established result that the superfluid weight is proportional to the integral of the quantum metric over the Brillouin zone**. This result is valid only for sufficiently small values of the Hubbard coupling U, which should be no larger the band gap separating the partially filled flat band from other fully filled or empty bands. I would expect that the same restrictions hold also in the case of the results of Ref. 31, 32 regarding the coherence length, although I have not investigated the matter very carefully.

**Answer:**
We fully agree with the fact that the superfluid weight is proportional to the integral of the quantum metric in the weak coupling regime (larger than the gap between the dispersive bands and the flat band), but we disagree on the fact that this can serve as a benchmark to anticipate the behavior of the coherence length. There is no direct connection between the superfluid weight and the coherence length.

**Referee**:
4- For instance it is shown both in Fig. 3b and Fig. 4b that for large enough values of U the coherence length, extracted from the decay of the anomalous correlation function, becomes smaller than the lower bound given by the quantum metric derived in Ref. 31,

32. This is not surprising in my opinion as this occurs for values of U that are comparable or larger than the band gap (the authors should check this), namely a regime where the coherence length is not controlled by the properties of a single partially filled band but also by other bands that are close in energy.

**Answer:**
We do not know if this is or not surprising, but, yes, this is indeed what is found in our calculations, that for large U the coherence length becomes smaller that the quantum metric.

**Referee**:
5- For very large values of U, a real space picture becomes more appropriate than a momentum space picture since a strong attractive Hubbard interaction leads to the formation of Cooper pairs in which both particles sit with high probability on the same lattice site. In summary, the numerical evidence provided in the manuscript regarding the stub lattice and the sawtooth lattice (Figs. 3 and 4) does not support the claim by the authors that the coherence length is disconnected from the quantum metric since this conclusion is drawn from the data for values of U comparable to or larger than the band gap, that is in a regime where a simple relation between quantum metric and coherence length is a priori not expected. The authors should carefully investigate the limits of validity of the results derived in Refs. 31 and 32 and, based on their findings, reassess their interpretation of the numerical data for the stub and sawtooth lattices.

**Answer:**
We do not completely understand this remark which seems somehow connected to the previous one. However the results depicted in Fig.3 and 4 are corresponding to weak, intermediate and strong coupling. As said before the data show in the limit of small U that there is no connection between the calculated coherence length and the quantum metric. For very large values of U one finds what one would expect: a coherence length smaller than the lattice parameter, which is the case in both figures (3 and 4).
We have investigated various situations, uniform and non uniform pairings, long range and short range hoppings, bipartite and non bipartite lattices, weak, intermediate and strong coupling. In all the cases investigated, we did not find any direct relation between the coherence length and the quantum metric (Ref.31, Ref.32).

**Referee**:
6 - The Creutz lattice, and also to some extent the χ-lattice, is peculiar since the single-particle correlation functions (both normal and anomalous) become zero for distances larger than one lattice spacing. First of all, I would like to point out that the vanishing of single-particle correlation functions of the Creutz ladder beyond a finite distance is not a surprising result, as stated by the authors, rather it is a straightforward consequence of the local integrals of motion of the Creutz ladder with an Hubbard

interaction, which have been found in [M. Tovmasyan, et al. Phys. Rev. B 98, 134513 (2018)].

**Answer:**

The referee says "*First of all, I would like to point out that the vanishing of single-particle correlation functions of the Creutz ladder beyond a finite distance is **not a surprising result**, as stated by the authors, rather it is a straightforward consequence of the local integrals of motion of the Creutz ladder with an Hubbard interaction*". In our view, these results are far from being trivial. In addition this remark means that the referee agrees with the fact that it is disconnected from the quantum metric although the conditions of large gap and uniform pairing are fulfilled! The prediction of Ref. 31 and 32 would be that the coherence length should be identical to the quantum metric. Regarding the fact that the referee says that this result was mentioned in "*M. Tovmasyan, et al. Phys. Rev. B 98, 134513 (2018)*". We are confused because we went through the manuscript, but we found no place where this result is pointed out. Regarding the chi-lattice the situation is different as discussed in the manuscript and depicted in Fig.6. But, as mentioned in our manuscript, the chi-lattice was especially studied in Ref.32 and our present results show that we fully disagree with their conclusion since they find that the coherence length is given by the quantum metric.

**Referee**:

7- Most importantly, the compact character of the correlation functions in the Creutz ladder implies that the coherence length, defined as the rate of the exponential decay of the correlation functions, is always zero regardless of the interaction strength. In my opinion, the Creutz ladder, rather then providing a counterexample to the statement that the coherence length is controlled by the quantum metric in some regime, shows that we should consider some other definition of the coherence length. A more appropriate definition in my opinion is to define the square of the coherence length as the average of the square of the distance between the two particles forming a Cooper pair. This is consistent with the role played by the quantum metric in providing a lower bound to the spread of Wannier functions, which is also defined as the average of the distance squared. Most importantly, this definition would give a nonzero and interaction-dependent value also in the case of the Creutz ladder. It might be that the results of Ref. 31 and 32 regarding the relation between quantum metric and coherence length refer to the latter definition, which is in my opinion more meaningful. The author should carefully check this point before drawing any conclusion.

**Answer:**

Even if the Creutz ladder is special in a sense, again as discussed in the previous points we did not consider the case of the Creutz ladder only. Part of the answer to this point is already discussed above. Regarding, the specific remark "*It might be that the results of Ref. 31 and 32 regarding the relation between quantum metric and coherence length refer to the latter definition*". We recall that our definition of the coherence length has

reproduced exactly the analytical BCS formula when the Fermi level was lying in a dispersive band. Regarding the results of Ref. 32, their correlation function $C(r_{ij})$ can be easily connected to ours ($K_{\alpha\beta}$ & $G_{\alpha\beta}$) through Wick's theorem.

$$C(r_{ij}) = \sum_{\alpha} \langle \hat{a}_{i\alpha\downarrow} \hat{a}_{i\alpha\uparrow} \hat{a}^{\dagger}_{j\alpha\uparrow} \hat{a}^{\dagger}_{j\alpha\downarrow} \rangle = (...) + \sum_{\alpha} |K_{\alpha\alpha}(0)|^2 + |G_{\alpha\alpha}(r_{ij})|^2$$

where "(...)" corresponds to constant ($r_{ij}$-independant) terms coming from anti-commutation relations. If we were using this correlation function to extract the coherence length, our conclusions would not change.

As said above, we again stress the fact that we have added in the manuscript a new appendix that explains the possible origin of the discrepancy between our findings and those of Law et al.

**Referee**:
8 - The case of the χ-lattice also shows that a more appropriate definition of the coherence length would be in terms of the average of the square of the interparticle distance. Indeed, the correlation functions are either zero at large distances if the orbital labels are equal, or they have an anomalous decay behavior if the orbital labels are different. While the definition of the coherence length in terms of the exponential decay length is not very useful in this case, because no good fit can be obtained in the case of different orbitals, the definition based on the distance square is and would probably give meaningful results. I would be interested to see how the coherence length, defined as the average of the distance squared, behaves in the case of the χ-lattice.

**Answer:**
Maybe there are other ways to define the coherence length but here we have decided to use the standard one, the one that leads in the conventional case (single band case) to the well known BCS expression as illustrated in Fig.2:

$$\xi = \frac{\hbar v_F}{\Delta}$$

**Referee**:
9- Note that the average involves also an average over all orbitals, therefore the different behaviors with respect to the orbital choice would not matter. Also, from the fact that the single-particle correlation functions vanish identically at any nonzero distance if the orbital indices are the same have prompted the author to state that "As a consequence, for any |U | the Cooper pair size is zero." In my opinion this statement is not justified because to evaluate the coherence length one should consider all possible orbital choices.

**Referee:**
10 - In the paragraph "Connection with recent studies" the authors state that: "Based on the fact that the coherence length is extracted from the long distance decay of $K_{\alpha\alpha}$ our findings clearly contradicts their prediction." The fact that the coherence length is extracted from the rate of the exponential decay at large distances is not obvious and the authors should check that this is in fact the same definition used in Refs. 31 and 32.

**Referee:**
11-As discussed above, other definitions are possible and probably more meaningful even in special cases such as the Creutz ladder and the χ-lattice. Also, there is no valid reason why the correlation function $K_{\alpha\beta}$ should be neglected for α different from β. For these reasons I do not agree with the discussion presented in this paragraph of the manuscript. To remedy this, it would be important for the authors to gain a more in depth understanding of the previous results of Ref. 31 and 32 in order to give a better interpretation of their numerical results.

**Referee:**
**12**- Whereas the manuscript is well written clear, few points require some attention:
- In the introduction it is written "The QM is connected to the real part of the quantum geometric tensor [19, 20] and provides a measure of the typical surface associated to the FB Bloch eigenstates". It is unclear for me what "measure of the typical surface associated to the FB Bloch eigenstates" means.

**Answer:**

First, we thank the referee for pointing out that our "manuscript is well written and clear". Indeed, we agree, it is a bit confusing to say *"measure of the typical surface associated to the FB Bloch eigenstates"* . We have modified this sentence in the introduction.

**Referee**:

**13**- At the beginning of the Theory and Method section: "On the other hand, the one-particle CF of the form ... always decays exponentially both in the superconducting phase and in the normal phase." In the absence of a gap, that is in the normal phase, the correlation function does not have an exponential decay behavior in general.

**Answer:**

We agree. To avoid any confusion, we have changed the sentence to *"On the other hand, the one-particle CF of the form ... always decays exponentially in the superconducting phase"* because here we focus on the superconducting phase only.

**Referee**:

**14**- In the section "Coherence length in dispersive bands" it is stated: "This density corresponds to the half-filling of the lower dispersive band." It would be useful to include a plot of the dispersion of the sawtooth lattice (as well as the stub lattice later on) to help the uninitiated reader understand this statement.

**Answer:**

Following the referee's demand, we have modified Fig.1 that now displays the dispersions in the 4 different lattices considered in our manuscript.

**Referee**:

**15**- In the conclusion: "It is found that the size of the Cooper pairs is less than one lattice spacing, both in the weak and strong coupling regime." This is a rather generic statement, which is for sure not applicable to all of the lattices studied in the manuscript. The authors should specify to which lattices they are referring to. The Conclusion should be significantly expanded in light of the above considerations.

**Answer:**

We agree, we have changed this sentence to: "It is found in this study that the size of the Cooper pairs is comparable to the lattice spacing, both in the weak and strong coupling regime." In addition, following the referee's recommendation and based on his remarks/questions and our answers, we have modified and expanded our conclusion a bit.