# Peer review of "Correlation functions and characteristic lengthscales in flat band superconductors"

_SciPost Physics_

## Round 1 · Referee Report · Anonymous (Referee 1) · 2024-9-19

Report

I thank the authors for their reply to all the questions of both referees and their adjusted manuscript. Unfortunately, the answers are unsatisfactory, so I cannot recommend publication. My main concern is that the authors did not consider several comments of both referees seriously. In particular, they did not attempt to resolve the raised issues but mostly insisted on statements in their previous manuscript. I will explain my evaluation in the following. I leave the question of whether further consideration for SciPost is reasonable to the editors. I go through the authors' replies in chronological order using their labeling.

Authors' reply to Referee 1

1 - At the point of resubmission of the manuscript on September 11th, the current version of arXiv:2308.05686 (Hu, Chen, and Law referred to as Ref 32) was version 5 (submission date July 14th) and not the considered version 3. Considering the most recent version would be fair when raising questions about its validity. Note that this manuscript is also still under review and might involve mistakes, which have been or will be corrected during this process. Indeed, Eq (S15), which the authors tried to recalculate, has been modified, given in (S36) in the new version, particularly at the essential step that the authors criticize in their reply and the new Appendix C. In contrast to the correlation function given in (L1) with both summations over alpha and beta, the authors of Ref 32 now consider only the correlation function summed over alpha, see (S29) in their version 5. The authors of Ref 32 justify this by assuming s-wave pairing before band projection. This different definition of the correlation function seems to be crucial. One would arrive at (L3') without that assumption. Instead, the authors of Ref 32 combine the coherence factors differently (which is possible due to the same index alpha) and obtain the form (L4) in combination with the so-called uniform pairing condition given in (S39) in version 5 of Ref 32.

I cannot judge whether this is the full line of thoughts required to compare the approaches or whether it was a "mistake in their calculation," as mentioned by the authors, that has now been corrected or clarified in the fifth version. However, this is not the essential point here. Instead of justifying their work by relating potential discrepancies to mistakes in others' work, I encourage the authors to provide evidence for the correctness of their own work. If the authors would like to indicate mistakes in others' work within their publication, I expect the highest standards, which, for instance, include using the latest version and considering all potential explanations that might also explain the discrepancy, including the possibility of different definitions of a coherence length (irrespectively whether the authors agree with such a definition or not).

2 (ii) - The authors' intention, as stated in the introduction, is "to address this issue [it is claimed that the coherence length can be expressed in terms of the BCS coherence length and the quantum metric] in several FB lattices and discuss our findings in connection with these predictions and with the existing literature". The authors now state that they "have decided to check their [those of Ref 32] expressions and found inconsistent results". In this sense, one might understand the authors' work as a direct reply to Ref [32]. My original question was intended to give the authors the opportunity to explain why the results in the previous literature seem unreasonable in the first place and to understand whether the authors have conjectures in mind that speak against the results of Ref [32]. This seems not to be the case, reading the authors' answer. They might clarify this point if this impression is not correct. It appears that the authors, instead, focus on verifying the concrete results of Ref [32]. In this case, using identical assumptions and discussing the exact details is necessary. However, the authors have not performed such an analysis; see comment above. Furthermore, I wonder if a direct reply to another paper that only intends to correct the results therein is worth publishing in SciPost as a new independent publication. I expect at least a detailed and clearly stated comparison between the work of Ref 32 and the approach of the authors. It should provide evidence for one or the other option that (partially, if too hard) answers the question of discrepancy and suggests a path toward a full solution.

2 (iii) - The others should link the four criteria directly to the models and state that explicitly.

2 (iv) - The question intended to ask the authors to explain the physics behind the observed same behavior when changing from the BCS to the BEC regime (red line in Fig 2). This question is in line with the questions raised by Referee 2.

Authors' reply to Referee 2

1 - The authors state, "Based on this result, it is natural to use the function K_{\alpha \beta} to extract the coherence length, which is done in the present work". The authors have seemingly proven that this definition of the coherence length is independent of the quantum metric. Considering the other comments of Referee 2, I think one has to interpret the referee's comment about the "relation between coherence length and quantum metric" such that the proper definition of coherence length is critical in this debate. I do not judge whether one or the other definition is more reasonable. I criticize the authors for not attempting to solve this issue but insisting on their definition. When performing a proper critique of the result of Ref [32], it is required, first, to use the exact definition of Ref [32] and show that this is not connected to the quantum metric or confirm that it is related to the quantum metric, second, to provide a discussion about pros and cons of different coherence length definitions and, third, provide a detailed study of the physics that is captured by which definition. I would consider this a constructive scientific study, and I understand Referee 2's questions in such a way that he requested such a study, which the authors unfortunately did not provide.

2 - As pointed out above, the key (and also interesting) question at this point is not whether the coherence length considered by the authors is independent of the quantum metric. The question is whether the coherence length, as defined by Ref. [32], is independent of the quantum metric. The authors seemingly insist that they answered that question, but, as elaborated in detail above, it is not correct. If the coherence length defined in Ref 32 does not depend on the quantum metric, where is the mistake in Ref. [32] (see my first point above)? If yes, why are the two definitions different, and which one is adequate for which physical question we want to answer with the concept of coherence length?

3 - In light of a coherence length definition involving the quantum metric, it is a valid question whether this length shows different behavior for U smaller or larger than the band gap. Referee 2 made the conjecture based on the experience with the superfluid stiffness. Since there is no direct connection, it remains a research question that the referee asked the authors to investigate. The question was not about a benchmark in the sense of a technical check.

4 - I encourage the authors to answer questions from referees such that it will help them understand the physics described in the manuscript.

6 to 11 - These comments and questions refer again to different definitions of the coherence length, where the referee requested to discuss other possibilities ("A more appropriate definition in my opinion ...", "a more appropriate definition"), but the authors insistent on their definition without further justification beyond those given before ("reproduced exactly the analytical BCS formula," "the standard one"). I would have expected the authors to consider the inconsistencies and arguments to elaborate more on different definitions of the coherence length.

Recommendation

Ask for major revision

---

## Round 1 · Referee Report · Sebastiano Peotta (Referee 2) · 2024-10-3

Strengths

1- The goal of the paper is to investigate the coherence length in lattice models with flat bands. The coherence length is an important observable quantity in superconductors and it is well-known and characterized in the case of dispersive bands, that is bands with bandwidth much larger than the superconducting gap. On the other hand, in the opposite flat band regime in which the superconducting gap dominates the bandwidth much less in known. This question has become topical with the discovery of new superconducting materials that approximate well the flat band limit, in particular magic angle twisted bilayer graphene, which features quasi-flat bands in the electronic band structure. Therefore this work is very timely.
2- The numerical calculations are done using mean-field theory for superconducting systems and appear to be correct. In particular they match closely results obtained in previous works using unbaised numerical methods.

Weaknesses

1- According to the authors, the main result of their work is that, based on their numerical and analytic results, a relation between the coherence length and the quantum metric derived in a previous work (Ref. 32 in the last version of the manuscript) is invalid. However, I do not see how their numerical results can be used to argue against the results of Ref. 32, in contrast to the authors's statement. 2- The authors define the coherence length as the length scale extracted from the exponential decay of a two-particle correlation function. Then, they extract numerically the coherence length in the case of the stub and sawtooth lattices. They find that the coherence length is in these cases larger that the quantum metric-based estimate of Ref. 32 for not too large values of the interaction strength $U$ (comparable or smaller than the band gap $E_{\rm gap}$). My interpretation of these results is that the lower bound obtained in Ref. 32 holds.
3- For large values of $U$ in the same lattices, the coherence length is smaller than the lower bound provided by the quantum metric. However, this fact cannot be used to argue against the quantum metric bound since this result of Ref. 32 is obtained by projecting on the relevant flat band. This is an approximation valid only if the interaction strength $U$ is sufficiently small compared to the band gap $E_{\rm gap}$. Therefore the results for large $U$ in the case of the stub and sawtooth lattices cannot be used to argue against the quantum metric-based bound of Ref. 32 since this is a regime in which the bound is not expected to hold. 4- In the case of the Creutz and $\chi$-lattice the authors find that the coherence length cannot be obtained from an exponentially fit since the relevant correlation function is exactly zero beyond few lattice sites in the case of the Creutz ladder and the decay behavior in the $\chi $-lattice is different from exponential. However, these results do not invalidate the relation between the quantum metric and the coherence length obtained in Ref. 32. The reason is that the authors of Ref. 32 define the square of the coherence length as the second moment of the Fourier transform of a correlation function. This means that the coherence length corresponds to the spread of the correlation function (in the same sense of Wannier function spread closely related to the quantum metric), which is a definition that can be applied also in the case when the decay behavior is not exponential. Personally, I prefer the definition of coherence length as proposed in Ref. 32, become it is useful also in the case of the Creutz and $\chi$-lattice. These are interesting lattices in which a superconducting state forms in the presence of an attractive interaction. In particular in the case of the $\chi$-lattice the observable properties of the superconducting state change with the parameter $\chi$, in particular the coherence length is expected to change, and the definition based on the spread of the correlation function captures this behavior. 5- In the latest version of the manuscript, the authors make a point that a step of a derivation in Ref. 32 is incorrect. In their argument they refer to the version 3 of Ref. 32, which is at present an arXiv preprint. However, in the latest version of Ref. 32 (currently 5), the relevant equations pointed out by the authors have been changed (see Eq. S36 in Ref. 32) and therefore Appendix C in the new version of the manuscript is not anymore relevant.

Report

In my opinion, the manuscript should not be published in the present form notwithstanding the validity of the numerical results presented therein. The main reason is that in my opinion the numerical results do not prove the main claim of this work, namely that the relation between quantum metric and coherence length, first proposed in Ref. 32, does not hold. On the other hand, it would be more fair in my opinion to state that the numerical results presented by the authors are in fact compatible with some sort of relation between quantum metric and coherence length. Moreover, I think it is premature to discuss the validity of the results of Ref. 32, which is still probably going through a lengthy review process given that five different versions have been posted so far in the arXiv. Before making strong statements regarding the validity of Ref. 32, the authors should wait until the final published version appears.

Requested changes

1- In the light of the previous discussion, the authors should thoroughly review and modify their conclusions. 2- The author should be point out that their definition of coherence length, based on an exponential fit of some correlation function, is different from the one used in Ref. 32, which corresponds to the spread of the same (or a similar) correlation function. Indeed, the authors of Ref. 32 use the following definition

$$ \xi^2 = -\frac{1}{2\mathcal{M}(\mathbf{0})} \frac{d^2\mathcal{M}(\mathbf{q})}{dq^2} $$
where
$$ \mathcal{M}(\mathbf{q}) = \sum_{ij\alpha} e^{i\mathbf{q}\cdot (\mathbf{r}i-\mathbf{r}_j)} \langle a \rangle. $$} a_{i\alpha\uparrow} a^\dagger_{j\alpha\uparrow} a^\dagger_{j\alpha\downarrow
By performing the second derivative in the first equations (or more rigorously by taking the trace of the matrix of second derivatives with respect to $q_i$) one obtains
$$ \xi^2 \propto \sum_{ij\alpha} |\mathbf{r}i-\mathbf{r}_j|^2\langle a \rangle, $$} a_{i\alpha\uparrow} a^\dagger_{j\alpha\uparrow} a^\dagger_{j\alpha\downarrow
which is precisely the second moment (or spread) of the correlation function. This is quantity is nonzero even in the case of a correlation function that vanishes identically beyond a finite range as in the case of the Creutz ladder, whereas the coherence length is strictly zero if one insists on defining it as the decay length in an exponential fit. 3- Neither in Ref. 32 nor in the present work it is clear why a specific correlation function should be chosen in order to compute the coherence length. A good answer to this question has been given in my opinion in [K. Higuchi, M. Higuchi, Journal of Physics Communications 5, 095003 (2021)], in which the coherence length is given by the spread (understood in the above sense) of the Cooper pair wavefunction, where the Cooper pair wave function is obtain as the eigenstate with the largest eigenvalue of the 2-body reduced density matrix. This definition is consistent with superconductivity being a manifestation of condensation of a macroscopic number of particles on the same two-body state. This reference is relevant and should be added among the citations and also discussed to some extent. Note in fact that the definition of coherence length used in this work recovers the standard BCS result $\xi = \hbar v_F/\Delta$ up to constants in the weak-coupling/dispersive band limit (see Eq. 25). 4- The authors should cite the work [M. Tovmasyan et al., Physical Review B 98, 134513 (2018)]. In this work it is shown that the Creutz ladder possesses some local symmetries that imply that the single-particle propagator is vanishing beyond a finite range. This important point is mentioned even in the abstract: "These conserved quantities are associated to the occupation of localized single quasiparticle states and force the single-particle propagator to vanish beyond a finite range." and discussed in several points throughout the text. In fact, the vanishing of the single-particle propagator is a result mentioned originally in Ref. 53 within the above work without proof. It is true that a proof of this fact a cannot be found in literature, as pointed out by the authors, but it is on the other hand rather straightforward. It is convenient to consider the basis of Wannier functions and the associated operator fermionic operators $w^\dagger_{n,i}$ labelled by a unit cell index $i$ and by the band index $n$. These are the operators that create one particle in one of the four-site Aharonov-Bohm cages of the Creutz ladder. The local symmetries of the interacting Creutz ladder can be written as
$$ C_i = C_i^\dagger = \exp[i\pi(w_{1,i}^\dagger w_{1,i}+w_{2,i}^\dagger w_{2,i}) ] $$
and have the property
$$ C_iw_{n,j}C_i = (-1)^{\delta_{i,j}}w_{n,j}. $$
Then, consider the single particle propagator $\langle \Phi|w_{m,i}w_{n,j}^\dagger |\Phi\rangle$ where $|\Phi\rangle$ is an eigenstate of all the $C_i$, for instance the ground state (it is assumed that the symmetries $C_i$ are not spontaneously broken). Since $C_i^2=1$ the eigenvalues of the symmetry operators are only $\pm1$ and one has that
$$ \langle \Phi|w_{m,i}w_{n,j}^\dagger |\Phi\rangle = \langle \Phi|C_lw_{m,i}w_{n,j}^\dagger C_l |\Phi\rangle = (-1)^{\delta_{i,l}}(-1)^{\delta_{j,l}}\langle \Phi|w_{m,i}w_{n,j}^\dagger |\Phi\rangle $$
if we take for instance $l = i\neq j$, the above equation implies that $\langle \Phi|w_{m,i}w_{n,j}^\dagger |\Phi\rangle =0$. The vanishing of the single particle propagator in the Wannier basis implies that also the single-particle propagator in the site basis vanishes beyond a finite range if the Wannier functions are compact as in the case of the Creutz ladder. If the Wannier functions are simply exponentially decaying, the single-particle propagator in the site basis is also exponentially decaying , but in general nonzero even at large distance. The above derivation is immediately extended to the time-dependent propagator since the many-body Hamiltonian commutes with the $C_i$.

Recommendation

Ask for major revision

---

## Round 1 · List of Changes

• Fig 1 has been modified (request of referee 2)
  • Fig 5 and Fig 6 have been modified (request of referee 1)
  • Conclusion has been extended (request of referee 2)
  • A new Appendix has been added (appendix C)
  • Paragraph before the conclusion has been modified
  • Typos and double labels have been corrected (request of referee 1)
  • Some abbreviations have been removed (request of referee 1)
  • Some sentences have been added to the introduction (request of referee 1 and 2)

---

## Round 2 · Referee Report · Anonymous (Referee 1) · 2024-10-28

Report

The authors of the manuscript provided satisfactory answers to the previous referee reports. Only a few minor presentational issues (summarized below) that the authors should clarify hinder me from recommending the publication of the manuscript.

Requested changes

1- The current abstract does not reflect the findings adjusted during the review process. I encourage the authors to highlight their analysis more precisely and extend it beyond the last sentence in its current form. 2- I thank the authors for adding Table 1 as a summary of the models. Unfortunately, the current position is not optimal, and a few further explanatory words are needed for the mentioned concepts (What does tunable QM precisely mean? What is uniform pairing?). Furthermore, a few more words at the end of the introduction would help to give a motivation why the five properties are relevant for the following discussion. 3- I thank the authors for clarifying the BCS-BEC crossover in Fig 2. Unfortunately, the presentation of Fig 2. is still misleading. It suggests that the crossover scale is at 0.04 (gray area), although the scale is around 1. I suggest removing the red coloring or extending the x-axis of this plot such that, indeed, only the BEC regime is indicated by the red color. 4- The authors connect the property of being bipartite to the existence of an FB at zero energy above Eq. (5). However, the chi-lattice does not show a zero mode although being bipartite. The authors may clarify this confusion. 5- The authors should clarify what they mean by "without changing the nature of the compact localized eigenstates." 6- The authors should clarify that "one would expect Delta_avg^{-1} from a standard BCS analysis" for \tilde \xi (see paragraph below Eq. (13)), where the authors potentially refer to the 1/Delta dependence of xi^(K).

Recommendation

Ask for minor revision

---

## Round 2 · Referee Report · Sebastiano Peotta (Referee 2) · 2024-11-6

Strengths

1- In the new version of the manuscript the authors consider an alternative definition of the coherence length, namely the one used in Ref. 32 [J.-X. Hu, S. A. Chen, and K. T. Law, arXiv:2308.05686]. In their analysis the authors show that this definition is not without problems and fails to give sensible results in the case of some lattice models.

Weaknesses

1- The authors claim in the abstract that the size of the Cooper pair is disconnected from the quantum metric. On the other hand, in the conclusion it is written "Nevertheless, ⟨g⟩ [the quantum metric] provides the correct order of magnitude in the sawtooth chain and for the stub lattice." I agree with the other reference that the abstract and conclusion should provide a more balanced description of the results obtained in the paper and they should also be consistent with each other. 2- Whereas the authors have used the definition of coherence length provided in Ref. 32 in the new version of the manuscript, this is only one of the possible options. As I pointed out in my previous report, the best definition of coherence length is in my opinion the one of Ref. 39, properly extended to multiorbital lattice models. This reference is cited in the new version but its results are mentioned only briefly. On the other hand, I think that this paper is rather important and the authors should analyze what happens when the definition of coherence length of Ref. 39 is used.

Report

I appreciate the fact the authors have extended their analysis and used the same definition of coherence length of Ref. 32, namely the reference claiming that there is a relation between quantum metric (integrated over the Brillouin zone) and coherence length. As shown in the new Eq. 11, this definition of coherence length amounts to the spread of the normal correlation function $G_{\lambda\lambda}(\mathbb{r})$. Interestingly, they highlight the fact that with this definition, the coherence length is zero in the stub lattice, the Lieb lattice and the $\chi$-lattice. Moreover, with the same definition it is found that the coherence length scales as $1/\sqrt{\Delta}$ and not $1/\Delta$, which is the usual result of BCS theory. In my opinion,this is a rather strong evidence that the results of Ref. 32 should be revisited.

On the other hand, these arguments are not sufficient to claim that the coherence length is completely disconnected from the quantum metric, as written in the abstract and in the conclusion. Indeed, the problems encountered with the definition of Ref. 32 might be easily resolved by using instead the definition of Ref. 39, which in the notation of the manuscript would read
\[
\xi^2 = \frac{\sum_{\lambda\eta} |\mathrm{r}|^2|K_{\lambda\eta}(r)|^2}{\sum_{\lambda\eta} |K_{\lambda\eta}(r)|^2}\,.
\]
Note that instead of the normal correlation function $G_{\lambda\eta}(\mathrm{r})$, the anomalous correlation function $K_{\lambda\eta}(\mathrm{r})$ appears. Moreover, the sum is carried out over all possible orbital pairs $\lambda\eta$ and not just for $\lambda=\eta$. Thus, the exact results in Eqs. 5 and 9 do not apply and the coherence length is finite in the case the stub lattice, the Lieb lattice and the $\chi$-lattice. Moreover, it has been shown in Ref. 39 that this definition reproduces the result of BCS theory for a dispersive band
\[
\xi = \frac{\hbar v_{\rm F}}{2\sqrt{2}\Delta}\,.
\]
Note the prefactor $1/(2\sqrt{2})$ which is ignored in the manuscript, for instance in Figure 7. The fact that one should sum over all possible orbital pairs probably modifies the numerical values of the coherence length shown in Fig. 7, maybe leading to a better agreement with the QM result.
From a conceptual point of view, it also seems more sensible that the coherence length should be calculated from the anomalous correlation function, which is the order parameter of the superconducting state.

Concluding, it seems to me that the definition of coherence length of Ref. 39 is the most appropriate one and does lead to consistent results, whereas the one in Ref. 32 has some issues. This leaves the question of the relation between the coherence length and the quantum metric very much open at present since the analytical results of Ref. 32 should be revisited, in particular one should check whether the approximations employed in this latter reference are indeed justified.

Requested changes

1- (Optional) Study what happens when the definition of coherence length of Ref. 39 is applied to the various lattice models considered in the manuscript. 2- Revisit abstract and conclusion to provide a balanced picture of the results. The claim that the coherence length is disconnected from the quantum metric is very strong and does not reflect accurately the results. In particular, in order to make such a claim it is necessary to consider all possible reasonable definitions of coherence length found in the literature, but at present this is not done in the current version of the manuscript since the definition provided in Ref. 39 is not taken into account. 3- Minor change: in the introduction the sentence “its square root provides a measure of the typical spread of the FB Bloch eigenstates.” should be modified. It is more accurate to say that the QM measures the minimal spread of the Wannier functions. 4- Minor change: after equation 4 "Note that a similar quantity have been used in Ref. [39] to extract the the Cooper pair size in conventional superconductors within the exact two-body problem." To my understanding the results of Ref. 39 are obtained withint the usual mean-field BCS approximation not by the solution of the two-body problem. Modify the sentence. 5- Minor change: Above Eq. 8 “Surprisingly, it is found that there are only two non-vanishing values corresponding respectively to |r| = 0 and a.” This is not surprising in view of the results of Ref. 43. Modify the sentence.

Recommendation

Ask for major revision

---

## Round 2 · Author Response

The PDF file of the response to the referees has been merged with the new PDF version of the manuscript.

---

## Round 2 · List of Changes

• Appendix C has been replaced
  • Section "Connection with recent studies" have been entirely rewritten
  • A new figure has been added in the main text (Fig. 7)
  • A new figure has been added in the new version of the appendix C (Fig. 8)
  • Changes in the text appear in blue for this second round. We recall that the changes in red correspond to the first round.
  • We have added three new references

---

## Round 3 · Referee Report · Sebastiano Peotta (Referee 2) · 2024-11-29

Strengths

1 - This work presents interesting results regarding the coherence length in various lattices
2 - The results are both numerical and analytical

Weaknesses

1 - Despite the extensive changes, some conclusions are still questionable,
for instance the claim that the Cooper pair size is zero in the $\chi$ lattice. This is a result of the fact that only correlation functions within the same sublattice are considered when making this claim, while correlation functions $G_{\lambda \eta}(r)$ and $K_{\lambda \eta}(r)$ for $\lambda \neq \eta$ are ignored. There is no valid reason to do this and according to some definitions the Cooper size is clearly nonzero. In any case, I hope this work will stimulate further discussions on the topic and possibly some consensus regarding a general definition of the Cooper pair size can be reached.

Report

The manuscript can now be published in my opinion, provided that the minor changes listed below are taken into consideration

Requested changes

1 - In the abstract and in the main text the authors talk about Green's function. Since the time variable is absent, it is more correct to talk about generic one-particle correlation functions, or more precisely the one-body density matrix, which in the case of superconducting systems contains a normal and an anomalous part
2 - Remove spaces before the colon symbol ":" in the abstract and in the text.
3 - At the beginning of the section on the sawtooth ladder it is written "The superconductivity in the "stub lattice" has been addressed in details in Ref. [38". I think that stub lattice should be changed with" sawtooth ladder" here

Recommendation

Publish (meets expectations and criteria for this Journal)

---

## Round 3 · Referee Report · Anonymous (Referee 1) · 2024-12-2

Strengths

The authors answered all questions of both referees satisfactorily and adjusted the manuscript accordingly.

Report

I can recommend publication when the minor changes requested by the second referee are fulfilled.

Requested changes

I agree with the three minor changes the second referee requested and ask the authors to adjust their manuscript accordingly.

Recommendation

Publish (meets expectations and criteria for this Journal)

---

## Round 3 · Author Response

The PDF file of the response to the referees has been merged with the new PDF version of the manuscript. All modifications regarding this 3rd round appear in green.

---

## Round 3 · List of Changes

• Abstract has been modified and rewritten
  • Table 1 has been moved, and the caption has been extended to define “tunable QM” and “uniform pairings”
  • Few sentences have been modified according to referee’s 2 suggestions
  • Figure 2 has been modified according to referee's 1 suggestion
  • Conclusion has been modified

---

## Editorial Decision

resubmitted